# Replication stress-inducing ELF3 upregulation promotes BRCA1-deficient breast tumorigenesis in luminal progenitors

Jiadong Zhou[1†], Xiao Albert Zhou[1*†], Li Hu[2†], Yujie Ma[1], Jun Zhan[3], Zhanzhan Xu[1], Mei Zhou[1], Qinjian Shen[1], Zhaofei Liu[4], Shaohua Ma[5*], Yuntao Xie[2*], Jiadong Wang[1,6*]

[1]Department of Radiation Medicine, School of Basic Medical Sciences, Peking University International Cancer Institute, Institute of Advanced Clinical Medicine, State Key Laboratory of Molecular Oncology, Peking University Health Science Center, Beijing, China; [2]Familial and Hereditary Cancer Center, Peking University Cancer Hospital and Institute, Beijing, China; [3]Department of Anatomy, Histology and Embryology, School of Basic Medical Sciences, Peking University Health Science Center, Beijing, China; [4]Medical Isotopes Research Center, Department of Radiation Medicine, School of Basic Medical Sciences, Peking University Health Science Center, Beijing, China; [5]State Key Laboratory of Molecular Oncology, Beijing, Key Laboratory of Carcinogenesis and Translational Research, Department of Thoracic Surgery I, Peking University Cancer Hospital and Institute, Beijing, China; [6]Department of Gastrointestinal Translational Research, Peking University Cancer Hospital, Beijing, China

*For correspondence:
xiaozhou@pku.edu.cn (XAZ);
doctor_msh@bjmu.edu.cn (SM);
zlxyt2@bjmu.edu.cn (YX);
wangjiadong1980@hotmail.com (JW)

†These authors contributed equally to this work

## eLife Assessment

In this **fundamental** study, the authors describe ELF3 as a candidate driver of luminal progenitor transformation, such that its up-regulation during replicative stress conditions and in BRCA1 deficient cells may permit cell proliferation by suppressing genome instability. While the work is certainly of interest, the supporting data remain **incomplete** as luminal progenitor cells could not be isolated, which would be needed in order to definitively determine whether ELF3 is a driver of transformation in these cells. Overall this paper may offer insight into mechanisms by which BRCA1 deficiency fuels breast tumorigenesis.

**Abstract** BRCA1 is a critical tumor suppressor, mutations in which greatly increase risks for many tumors in carriers, most notably breast cancer. Luminal progenitor cells (LPs) are the currently recognized cells of origin of BRCA1-deficient breast cancers. However, the reason why LPs are prone to transform with BRCA1 deficiency has not been elucidated. Here, using single-cell sequencing of human *BRCA1* mutant breast cancers and RNA sequencing (RNA-seq) of *BRCA1*-deficient normal mammary cells, we reveal that replication stress is a feature of LPs and a driving factor during BRCA1-associated tumorigenesis. Mechanistically, replication stress and BRCA1 deficiency lead to significant upregulation of ELF3 expression. ELF3 can help suppress excessive genomic instability and promote LP transformation with BRCA1 deficiency. Moreover, ELF3 emerged as a core transcription factor regulating LP genes, leading to LP expansion. Our findings suggest that replication stress

is a driving factor during BRCA1-associated tumorigenesis in luminal progenitor cells and elucidates the key role of ELF3 during this process.

## Introduction

*BRCA1* is one of the most important tumor suppressor genes. Carriers of *BRCA1* germline mutations are at greatly increased risks for developing various tumors, including breast, ovarian, prostate, colon, pancreatic, and gastric cancers (*Ford et al., 1994*; *Friedenson, 2005*). Notably, BRCA1 is most closely associated with breast cancer. However, why BRCA1-associated tumors exhibit significant tissue specificity is unknown, and the effects of BRCA1 defects in normal human mammary cells are largely obscure.

In addition to the tissue specificity of tumors caused by BRCA1 deficiency, breast cancers associated with BRCA1 deficiency exhibit a set of other distinctive features. Most *BRCA1* mutation carriers develop triple-negative breast cancers (TNBCs), in which tumors show negative expression of estrogen receptor (ER), progesterone receptor (PR), and Her2 (human epidermal growth factor receptor 2), and these cancers are usually also classified as basal-like breast cancers. The currently recognized cells of origin of BRCA1-deficient breast cancers are the LPs in the mammary gland (*Lim et al., 2009*; *Molyneux et al., 2010*; *Nolan et al., 2016*; *Wang et al., 2019*). LPs show abnormal expansion in the normal mammary tissue of *BRCA1* mutation carriers (*Lim et al., 2009*; *Proia et al., 2011*), which might be caused by a differentiation blockade (*Hu et al., 2021*). Investigating the traits of LPs is important for tumor prevention and treatment in *BRCA1* mutation carriers and a wider range of patients with TNBCs and basal-like breast cancer. However, the reason LPs are prone to transform into cancer cells when BRCA1 is deficient is unclear.

ELF3 is a member of the ETS family of transcription factors. Early reports suggested that ELF3 may regulate the development and differentiation of mammary tissue (*Oliver et al., 2012*). Intriguingly, ELF3 has opposite functions in tumorigenesis in different tissues. In breast cancers, ELF3 shows high expression (*Chang et al., 1997*), plays an oncogenic role by promoting EMT (*Schedin et al., 2004*) and forms a positive feedback loop with Her2 to maintain the transformation phenotype of Her2+ breast cancers (*Neve et al., 2002*). However, the mechanism of high ELF3 expression in certain types of breast cancer and its association with breast cancers beyond the Her2+ subtype is currently unknown.

In this study, utilizing single-cell sequencing of *BRCA1* mutant breast cancers and time-resolved RNA-seq of *BRCA1*-deficient normal mammary cells, we revealed that LPs have a trait of higher replication stress compared to other cell populations in the normal mammary tissue, endowing them with the potential for transformation when BRCA1 is deficient, and ELF3 plays a key role in the process. Mechanistically, ELF3 expression is significantly upregulated by replication stress and BRCA1 deficiency, dependent on the ATR-Chk1-E2F axis and the transcriptional regulation function of BRCA1, respectively. In addition, ELF3 can help suppress excessive genomic instability, promoting the tolerance of LPs to BRCA1 deficiency. Notably, BRCA1 deficiency causes cells to display LP transcriptional profile characteristics, leading to dedifferentiation and expansion of LPs, and ELF3 is a core transcription factor during this process. Our study reveals why BRCA1 deficiency is prone to result in tumorigenesis in LPs and elucidates the key role of replication stress and ELF3 during this process, suggesting that ELF3 could be a promising target for BRCA1-associated breast cancers.

## Results

### LPs have higher levels of replication stress compared with other mammary cell populations

LPs in the mammary gland are the currently recognized cells of origin of BRCA1-deficient breast cancers. However, the reason for LPs are prone to transform with BRCA1 deficiency has not been elucidated. Firstly, we considered that LPs, which are stimulated by upstream progesterone paracrine signals through RANKL (*Nolan et al., 2016*), may have higher levels of replication stress, which can trigger tumorigenesis by increasing genomic instability. We utilized both published data and our published single-cell sequencing data (*Hu et al., 2021*) to address this question. We evaluated the

replication stress levels in different mammary cell populations from the dataset of *Lim et al., 2009* by scoring multiple DNA replication pathway gene expression as previously studied (*Dreyer et al., 2021*; *Takahashi et al., 2022*). The result showed that LPs have the highest level of replication stress among all subgroups of normal human mammary cells (*Figure 1A*, *Supplementary file 1*). We further analyze our previously obtained single-cell sequencing data of four *BRCA1* mutation carriers (case #1–4) and normal mammary cells of three matched non-carriers (case #5–7). According to our previous analysis, case #3 *BRCA1* mutant carrier developed basal-like breast cancer, which was derived from LPs. Importantly, in this case of *BRCA1* mutant carrier, single-cell sequencing data revealed that tumor cells had the highest levels of replication stress, and LPs had the second-highest levels (*Figure 1B*, *Figure 1—figure supplement 1A* and *Supplementary file 2*). Moreover, we applied Monocle to reconstruct the transformation trajectories from normal mammary cells to tumor cells in the *BRCA1* mutation carrier and found that the levels of replication stress showed an increasing trend during the process of LP transformation into tumor cells (*Figure 1C*, *Figure 1—figure supplement 1B*). These data demonstrate that replication stress is an important trait of LPs and might be a key driver of BRCA1-deficient tumorigenesis, which is consistent with the previous report that *BRCA1* mutant mammary epithelial cells are defective in replication stress suppression (*Pathania et al., 2014*).

Next, we aimed to find other key regulators during BRCA1-deficient tumorigenesis in LPs. In addition to inducing replication stress, BRCA1 deficiency also profoundly affects the transcriptional profile of normal mammary cells by disrupting the transcriptional function of BRCA1, which may further boost the process of tumorigenesis. To investigate the transcriptional impact of BRCA1 deficiency in normal human mammary epithelial cells, we took advantage of the Tet-on system to build an inducible knockdown model in MCF10A cells. By adjusting the treatment condition of DOX (doxycycline), different time points of BRCA1 deficiency can be controlled. Treatment with DOX causes a constant and efficient knockdown of BRCA1, verified at mRNA and protein levels (*Figure 1—figure supplement 1C*). We showed that DOX treatment induces significant growth arrest and decreased cell survival (*Figure 1—figure supplement 1D*) as well as an increase in genomic instability (*Figure 1—figure supplement 1E*), consistent with the phenotypes of BRCA1-deficient cells (*Zheng et al., 2000*), confirming that our system functioned as expected. We then performed time-resolved RNA-seq with different durations of DOX treatment (2, 5, and 10 days; *Figure 1—figure supplement 1F*). Notably, long-term DOX treatment resulted in a decreased number of colony formation, indicating impacted cell survival. Thus, cells collected for sequencing at the end of 10 days DOX treatment may have gained a survival advantage compared to cells that did not survive throughout the DOX treatment. Differential gene expression analysis revealed that BRCA1 was efficiently knocked down during the whole experiment (*Figure 1D*). Importantly, the principal component analysis (PCA) revealed that samples showed a time-course distribution in the dimension of PC1, which has the dominant power of explanation (*Figure 1E*, *Figure 1—figure supplement 1G*). These data demonstrate that the variations between samples can be largely attributed to time. To further investigate the modes of differentially expressed genes (DEGs) that change over time, we carried out the DEG cluster analysis using Mfuzz (*Kumar and E Futschik, 2007*). Analysis of the Day 10 DEGs showed that genes could be divided into five clusters with different expression change patterns (*Figure 1F*). We mainly focused on Cluster 1, for these sharply upregulated genes may play more important roles in cells gaining survival advantage with BRCA1 deficiency at Day 10, mimicking the initiation of BRCA1 deficiency-associated tumorigenesis. To find out the most potent regulator upregulated by both replication stress and BRCA1 deficiency, we performed RNA-seq in HU-treated MCF10A cells and overlapped the upregulated genes in HU RNA-seq and Cluster 1 genes in BRCA1 deficiency RNA-seq (*Figure 1G*). The results showed that ELF3, a transcription factor, is one of the most dominant genes in this analysis. ELF3 has been reported to be involved in functions, including EMT in human breast cancer and promotes the transformation of normal human mammary cells (*Schedin et al., 2004*; *Prescott et al., 2004*). Therefore, we speculated that ELF3 could be an important driving factor during the tumorigenesis of BRCA1-deficient and basal-like breast cancer.

## ELF3 is upregulated in BRCA1-associated breast cancer and is related to a worse prognosis

We next explored the expression of ELF3 in breast cancers in the TCGA and METABRIC databases to further investigate the association between ELF3 and BRCA1 and validate our previous analysis.

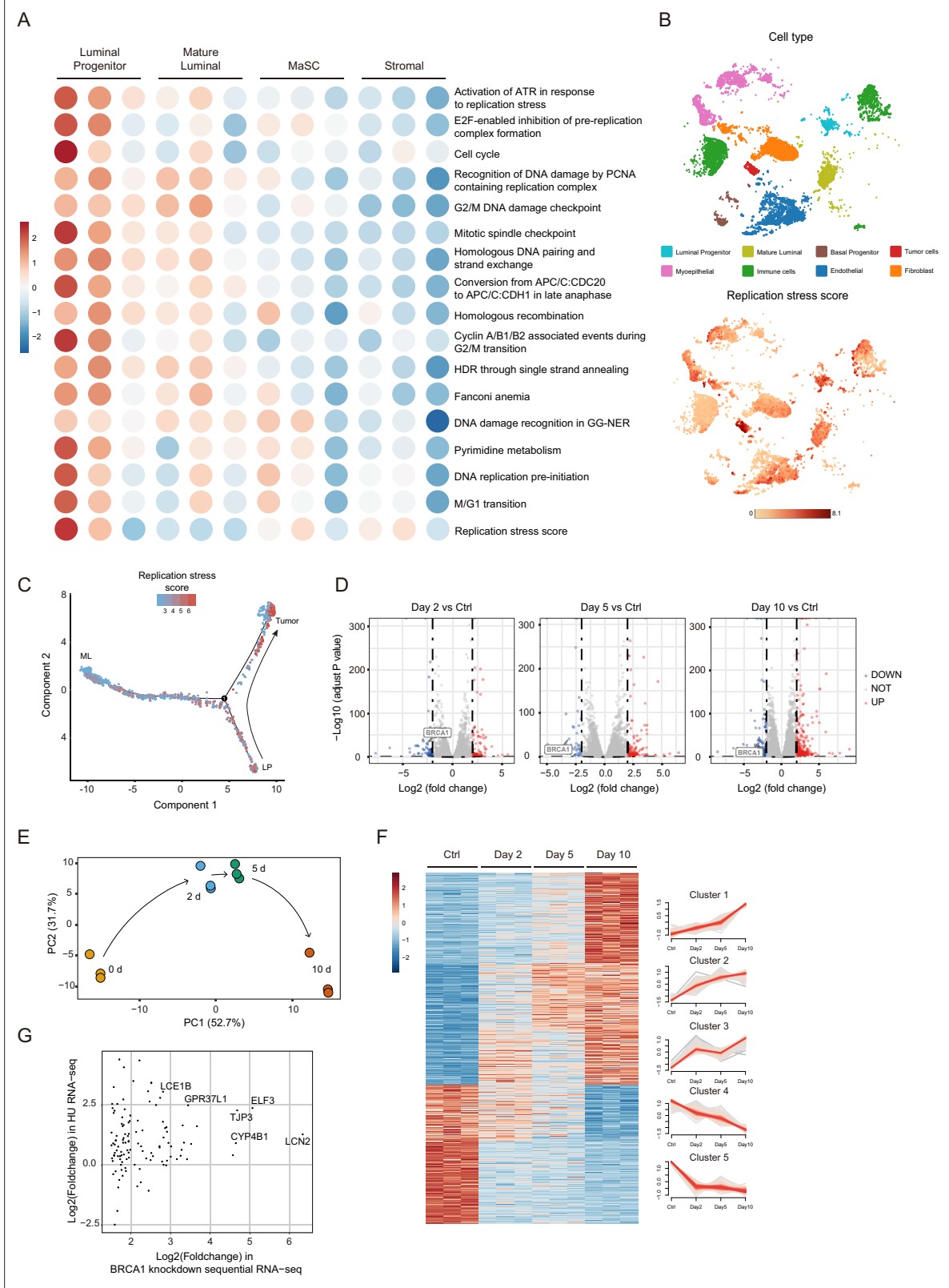

**Figure 1.** LPs have higher levels of replication stress compared with other mammary cell populations. (**A**) Replication stress pathway scores of different human normal mammary cell populations, scaled by row, transformed to mean = 0 and variance = 1. The gene expression data were from *Lim et al., 2009*. The gene expression signatures were from *Dreyer et al., 2021* and *Takahashi et al., 2022*. (**B**) T-distributed statistical neighbor embedding (t-SNE) plots visualizing the replication stress scores and ELF3 expression levels in breast cancer and normal mammary tissue from a *BRCA1*-mutated

*Figure 1 continued*

triple-negative breast cancer patient. The cells are colored by cell type (top) and replication stress score (bottom). (**C**) Monocle-generated cell trajectories visualizing the change in replication stress scores during tumorigenesis. The tumor cells and compared normal luminal cells were obtained from a triple-negative breast cancer patient (case #3) carrying a *BRCA1* germline mutation. LP, luminal progenitor cells; ML, mature luminal cells. (**D**) Volcano plots showing differentially expressed genes (DEGs) in MCF10A-shBRCA1-Tet-on cell RNA-seq results after 1 µg/mL doxycycline (DOX) treatment for 2 (left), 5 (middle), and 10 (right) days versus the Ctrl group. DEGs are genes with |log2-fold change|>2 and FDR <0.001. (**E**) PCA plot of MCF10A-shBRCA1-Tet-on cell RNA-seq results. (**F**) Mfuzz clustering analysis of Day 10 versus Ctrl DEGs. The heatmap (left) shows the differential expression mode of all genes in each cluster. The line chart (right) shows the global changing mode of each cluster. (**G**) Overlap of HU-treated MCF10A RNA-seq results and MCF10A-shBRCA1-Tet-on cell RNA-seq Cluster 1 genes.

The online version of this article includes the following source data and figure supplement(s) for figure 1:

**Figure supplement 1.** Luminal progenitor cells (LPs) have higher levels of replication stress compared with other mammary cell populations.

**Figure supplement 1—source data 1.** Original western blots for *Figure 1—figure supplement 1C*, indicating the relevant bands.

**Figure supplement 1—source data 2.** Original files for western blot analysis displayed in *Figure 1—figure supplement 1C*.

In both clinical databases, ELF3 expression is significantly higher in BRCA1-associated breast tumors than in non-BRCA1-associated breast tumors (*Figure 2A*). Moreover, the expression levels of BRCA1 and ELF3 showed a significant negative correlation, consistent with our findings that ELF3 was upregulated under BRCA1 deficiency (*Figure 2B*). Since most *BRCA1* mutation carriers develop basal-like breast cancers, we next investigated the association between ELF3 and basal-like breast cancers. We exploited the TCGA and METABRIC databases to compare ELF3 expression levels among different breast cancer subtypes. As we expected, ELF3 showed the highest expression levels in basal-like breast cancer in both databases (*Figure 2C*). We took advantage of novel subgroups identified by the METABRIC (*Curtis et al., 2012*) and further explored ELF3 expression in ten IntClusts of the METABRIC database. Among all subgroups, IntClust 5 was enriched with Her2+ cancers, and IntClust 10 was enriched with basal-like cancers. These two subgroups showed the highest expression levels of ELF3, with no significant difference between them (*Figure 2D*). The higher levels of ELF3 expression in Her2 subtype breast cancer are consistent with previous studies showing that ELF3 may be a downstream transcription target of Her2 and that there might be a positive feedback loop between them (*Neve et al., 2002*; *Eckel et al., 2003*; *Coppe et al., 2010*). On the other hand, IntClust 4 shows a favorable outcome and has a 'CNA-devoid' feature, which has the lowest level of ELF3 expression. These data are consistent with our findings that ELF3 is upregulated by BRCA1 loss, indicate that ELF3 may play an important role in basal-like breast tumorigenesis, and moreover, the expression of ELF3 may be related to genomic stability status. Furthermore, among human breast cancer tissue samples, ELF3 expression levels were significantly higher in TNBCs than in other subtypes (*Figure 2E and F*). Additionally, TNBCs have higher ELF3 expression levels than the Her2 subtype (*Figure 2E and F*), revealing a tighter connection between ELF3 and BRCA1-associated breast cancers.

ELF3 shows significant copy number amplification in breast cancer (*Oettgen et al., 1997*). Thus, we wondered whether the high expression levels of ELF3 in BRCA1-associated breast cancers are due to copy number variation. Therefore, we analyzed the proportion of *ELF3* copy number amplification in different subtypes (*Figure 2—figure supplement 1*) and found that this was not the case. In TCGA basal-like breast cancer, the *ELF3* amplification proportion was not higher than that in other subtypes but was only 3.5%, which was lower than the total level (7.3%). These data demonstrate that the upregulation of ELF3 in BRCA1-associated breast cancer is not caused by copy number amplification.

Furthermore, we investigated whether ELF3 expression could be a biomarker for breast cancer prognosis. We found that higher ELF3 expression is correlated with worse prognosis in TNBC and basal-like breast cancer (*Figure 2G*). In the Her2 subtype, higher ELF3 expression is also related to shorter overall survival, consistent with previous reports that as a Her2 target gene, ELF3 may help promote tumor invasion (*Coppe et al., 2010*). Interestingly, ELF3 expression shows less or no prognostic prediction value in luminal A and luminal B subtypes, suggesting that ELF3 plays a more powerful role in TNBC and basal-like breast cancer. These data imply that in BRCA1-associated breast cancer, ELF3 tends to be upregulated and plays a 'driver' role, promoting tumor progression and leading to a worse prognosis.

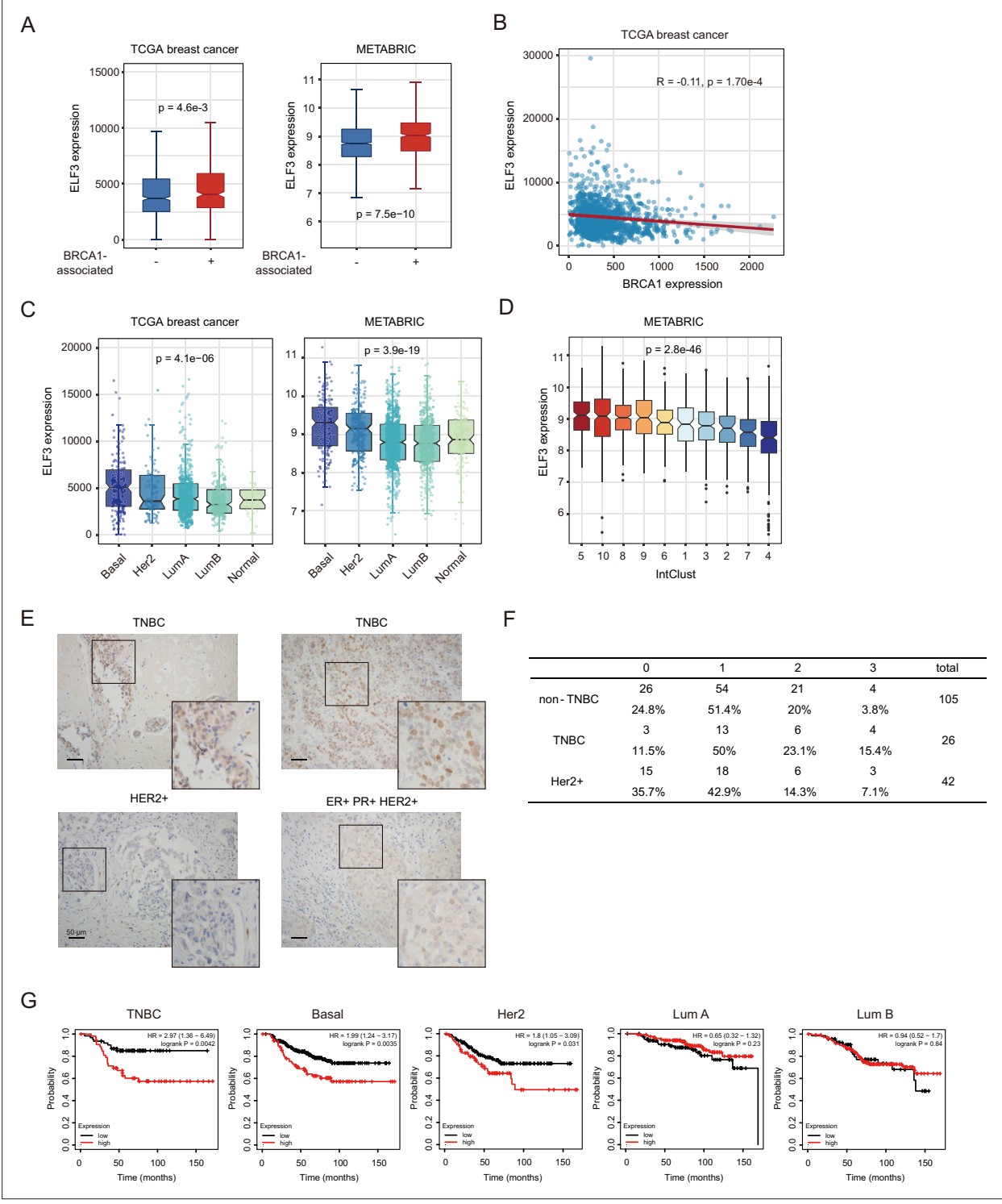

**Figure 2.** ELF3 is upregulated in BRCA1-associated breast cancers and is related to a worse prognosis. (**A**) ELF3 expression levels in BRCA1-associated breast cancers and BRCA1-non-associated breast cancers in the TCGA and METABRIC databases. BRCA1-associated breast cancers comprise samples with *BRCA1* mutations, heterozygous loss, and homozygous deletion (Mann–Whitney U test) (TCGA: BRCA1-associated, N=362; BRCA1-non-associated, N=720; METABRIC: BRCA1-associated, N=540, BRCA1-non-associated, N=1364). The bars (whiskers) in the boxplots represent the minimum and maximum values within the range of Q1-1.5*IQR and Q3+1.5*IQR, but do not include outliers. (**B**) Correlation of ELF3 and BRCA1 expression levels in TCGA breast cancer datasets. (**C**) ELF3 expression levels of different molecular subtypes of breast cancers in the TCGA and METABRIC databases (one-way ANOVA). (TCGA: Basal N=171, Her2 N=78, LumA N=499, LumB N=197, Normal N=36; METABRIC: Basal N=199, Her2 N=220, LumA N=679, LumB N=461, Normal N=140) (**D**) ELF3 expression levels of different IntClusts of breast cancers in the METABRIC database (one-way ANOVA; for

*Figure 2 continued on next page*

*Figure 2 continued*

IntClust 5 and 10, Mann–Whitney U test, ns, no significance; 5 N=184, 10 N=219, 8 N=289, 9 N=142, 6 N=84, 1 N=132, 3 N=282, 2 N=72, 7 N=182, 4 N=318). (**E**) Immunohistochemistry images of human triple-negative, Her2+, and ER+ PR+ Her2+ breast cancer samples stained with the ELF3 antibody. (**F**) ELF3 expression levels of the indicated types of human breast cancers (Mann-Whitney U test, *$p<0.05$). (**G**) Survival analysis of the indicated subtypes of human breast cancer in the KMplot database. For the TNBC group, samples with negative expression of all hormone receptors (estrogen receptor ER, progesterone receptor PR, and human epidermal growth factor receptor 2 Her2) are included. In the TNBC group, high expression (n=54), low expression (n=62); in the basal group, high expression (n=85), low expression (n=211); in the Lum A group, high expression (n=121), low expression (n=101); in the Lum B group, high expression (n=133), low expression (n=67); and in the Her2 group, high expression (n=72), low expression (n=126).

The online version of this article includes the following figure supplement(s) for figure 2:

**Figure supplement 1.** ELF3 expression upregulation in basal-like breast cancers is not due to amplification.

## ELF3 upregulation is induced by replication stress via the ATR-Chk1-E2F axis and by BRCA1 deficiency via GATA3 transcription

We next focused on the mechanisms of ELF3 upregulation in the presence of replication stress and BRCA1 deficiency. Consistent with the HU RNA-seq analysis, our single-cell RNA-seq data showed that in both *BRCA1* WT and mutant carriers, the levels of replication stress and ELF3 expression showed a significant positive correlation (***Figure 3A***). ELF3 was also upregulated by HU-induced replication stress in MCF10A cells (***Figure 3B***). Since ATR is the key kinase that responds to replication stress, we treated cells with ATRi and found that it could efficiently block ELF3 upregulation induced by HU, while ATMi had no such effect (***Figure 3B***). Moreover, downstream of ATR, ELF3 expression upregulation depended on Chk1 activation but not Chk2 activation (***Figure 3C***). It has been reported that the transcriptional activation induced by replication stress-ATR-Chk1 pathway largely depends on E2F transcription factor activation (***Bertoli et al., 2013***). Under replication stress, ATR-Chk1-mediated phosphorylation of E2F6 blocks its inhibitory function on E2F and activates E2F transcription. Thus, we overexpressed E2F6 and found that E2F6 largely blocks the upregulation of ELF3 under HU treatment conditions (***Figure 3D***), validating that ELF3 upregulation relies on E2F6 inactivation and E2F de-suppression. Moreover, utilizing ENCODE ChIP-seq data, we found that the *ELF3* promoter is indeed bound by E2F1 (***Figure 3E***). These data demonstrate that ELF3 is upregulated by the ATR-Chk1-E2F axis under replication stress conditions.

Then we investigate the mechanism of ELF3 upregulation induced by BRCA1 deficiency. ELF3 expression gradually increased during 10 days of DOX treatment (***Figure 3F***). ELF3 upregulation was further confirmed by siRNA-mediated BRCA1 knockdown in MCF10A cells to strictly exclude the unknown influence of the Tet-on system (***Figure 3G***). BRCA1 has transcription regulation functions by interacting with other transcription factors to bind to gene promoters and regulate gene expression (***Mullan et al., 2006***). Therefore, we explored transcription factors associated with BRCA1, including GATA3 (***Tkocz et al., 2012***) and Myc (***Kennedy et al., 2005***), for whether the binding motifs of these transcription factors appear in the *ELF3* promoter sequence utilizing the JASPAR database (***Figure 3—figure supplement 1A***). We found that E2F1 was present in the prediction list with a high relative score, consistent with our results that E2F1 can bind the *ELF3* promoter. The prediction results show that the motif of GATA3 appeared in the *ELF3* promoter multiple times with an overall high relative score. Indeed, knockdown of GATA3 in MCF10A cells also strongly induced ELF3 expression (***Figure 3H***). Similarly, we found multiple GATA3 binding peaks at the promoter of *ELF3* in two independent ChIP-seq datasets from the ENCODE database, consistent with two of the GATA3 binding sequences predicted in the JASPAR database (***Figure 3I***). These data suggest that *ELF3* is a GATA3 downstream target gene. Importantly, when both BRCA1 and GATA3 were knocked down, there was no significant additive effect on ELF3 upregulation (***Figure 3J***, ***Figure 3—figure supplement 1B***), implying that BRCA1 and GATA3 mainly act in the same pathway for ELF3 induction.

Notably, we found that BRCA1 knockdown and HU treatment have a superimposed effect on upregulating ELF3 expression levels (***Figure 3K***, ***Figure 3—figure supplement 1C***). Furthermore, GATA3 knockdown and HU treatment also had a significant superimposed effect in upregulating ELF3 expression (***Figure 3L***, ***Figure 3—figure supplement 1D***). These data validate that the BRCA1-GATA3 axis and replication stress regulate ELF3 expression independently.

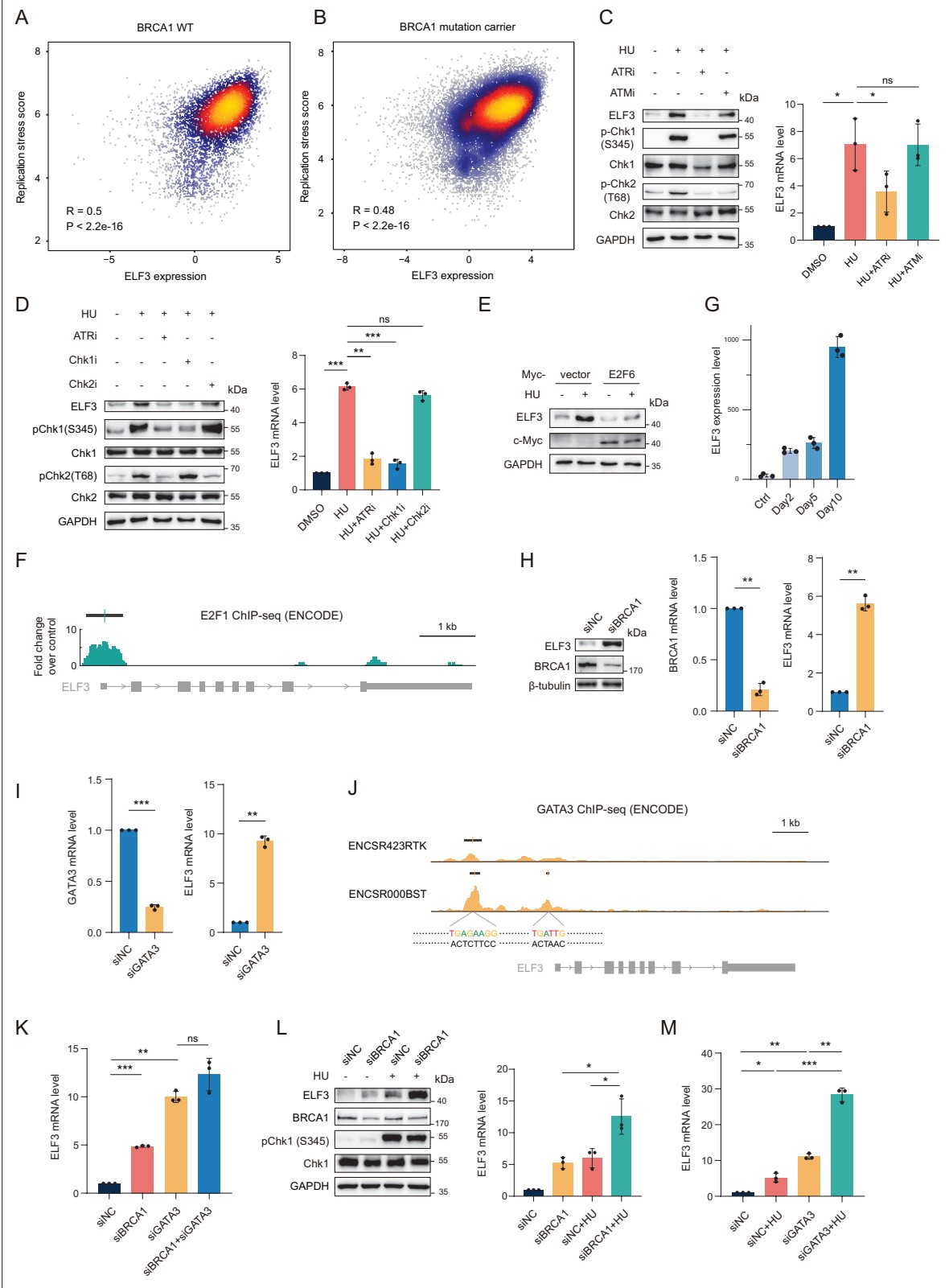

**Figure 3.** ELF3 upregulation is induced by replication stress via the ATR-Chk1-E2F axis and by BRCA1 deficiency via GATA3 transcription. (**A**) Spearman correlation analysis of ELF3 expression levels and replication stress scores in normal mammary tissue from non carriers (left) and *BRCA1* mutation carriers (right). Expression data is from single-cell RNAseq data, including three BRCA1 WT patients and four *BRCA1* mutant patients. All cells (normal cells and tumor cells) are involved, and ELF3 expression was normalized by reads in each cell. (**B**) Western blot and RT–qPCR results of MCF10A cells

*Figure 3 continued on next page*

*Figure 3 continued*

treated with the indicated drugs for 12 hr. Drug concentration: HU 1 mM, ATRi (VE821) 10 µM, ATMi (KU5593) 10 µM. (**C**) Western blot and RT–qPCR results of MCF10A cells treated with the indicated drugs for 12 hr. Drug concentrations: HU 1 mM, ATRi (VE821) 10 µM, Chk1i (GDC-0575) 50 µM, Chk2i (BML-277) 10 µM. (**D**) Western blot results of MCF10A cells transfected with the empty vector or Myc-E2F6 and treated with DMSO or 1 mM HU for 12 hr. (**E**) ChIP-Seq enrichment signal (green plot, ENCODE ENCFF858GLM; data are representative of two independent experiments) and the location of detected peaks (black line, ENCODE ENCFF692OYJ; irreproducible discovery rate (IDR) thresholded peaks) of E2F1 in MCF7 cells from the ENCODE database. (**F**) ELF3 expression changes as BRCA1 deficiency duration increases. ELF3 expression levels are indicated by the CPM of the RNA-seq results (means ± SD, n=3). (**G**) Western blot and RT–qPCR results of MCF10A cells transfected with siNC or siBRCA1 for 48 hr. (**H**) RT–qPCR results of MCF10A cells transfected with siNC or siGATA3 for 48 hr. (**I**) ChIP-Seq enrichment signal (yellow plot up, ENCODE ENCFF384CPN; yellow plot down, ENCODE ENCFF342GNN. Data are representative of two independent experiments) and the location of detected peaks (black line up, ENCODE ENCFF352QVM; black line down, ENCODE ENCFF437NQS; both are irreproducible discovery rate (IDR) thresholded peaks) of GATA3 in MCF7 cells from two independent studies (ENCSR423RTK and ENCSR000BST) from the ENCODE database. The motif sequences were obtained from the JASPAR database. (**J**) RT–qPCR results of MCF10A cells transfected with the indicated siRNA for 48 hr. (**K**) Western blot and RT–qPCR results of MCF10A cells transfected with siNC or siBRCA1 for 24 hr followed by treatment with DMSO or 1 mM HU for 12 hr. (**L**) RT–qPCR results of MCF10A cells transfected with siNC or siGATA3 for 24 h followed by treatment with 1 mM HU for 12 hr. In all panels of RT–qPCR results, data are presented as the mean ± SD, n=3, *$p<0.05$; **$p<0.01$; ***$p<0.001$; ns, no significance, by paired two-tailed Student's t-test.

The online version of this article includes the following source data and figure supplement(s) for figure 3:

**Source data 1.** Original western blots for *Figure 3C, D, E, H and L*, indicating the relevant bands.

**Source data 2.** Original files for western blot analysis displayed in *Figure 3C, D, E, H and L*.

**Figure supplement 1.** The replication stress response pathway is activated in BRCA1-associated breast cancers.

## ELF3 helps cells tolerate replication stress and sustain cell survival

Next, we investigated the impact of ELF3 upregulation under replication stress conditions and BRCA1 deficiency. Based on the association between ELF3 and the worse prognosis in BRCA1-associated breast cancer patients (*Figure 2G*), we hypothesized that upregulation of ELF3 could help cells deal with replication stress and suppress excessive genomic instability, facilitating cancer initiation and cancer evolution. As expected, ELF3 knockdown greatly increased the sensitivity of MCF10A cells to HU and cisplatin (*Figure 4A and B*, *Figure 4—figure supplement 1A*). This finding suggests that ELF3 can help normal cells deal with replication stress. Furthermore, ELF3 knockdown in HCC1937 and SUM149PT cells, both of which are human breast cancer cell lines with *BRCA1* mutations and allelic loss, also caused a significant decline in cell proliferation (*Figure 4C and D*, *Figure 4—figure supplement 1B and C*), suggesting that ELF3 is essential for *BRCA1* mutant breast cancer cell survival. Moreover, HCC1937 and SUM149PT cells became more sensitive to HU and cisplatin when ELF3 was deficient (*Figure 4E and F*, *Figure 4—figure supplement 1D and E*). Since PARP inhibitors can induce replication fork stalling by PARP1 trapping and lead to replication stress in BRCA1-deficient cells, we also tested olaparib sensitivity in HCC1937 cells and obtained similar results (*Figure 4—figure supplement 1F*). These results demonstrate that in *BRCA1* mutant breast cancer cells, ELF3 can help cells tolerate replication stress and sustain cell survival.

In addition, utilizing the CellMiner database, we found that among nearly sixty diverse human cancer cell lines, lower ELF3 expression is correlated with higher drug sensitivity to cisplatin and etoposide (*Figure 4G*). Consistent results were obtained using the GDSC database (*Figure 4H*, *Figure 4—figure supplement 1G*). Moreover, in nude mice, SUM149PT tumors with ELF3 deficiency were significantly smaller than those in the control group (*Figure 4I*). These data indicate that ELF3 may help normal cells tolerate replication stress and subsequent DNA damage, allowing transformation into malignant tumors. During cancer progression, *BRCA1* mutant cancer cells gradually become reliant on high ELF3 expression to preserve a certain degree of genomic stability, making ELF3 a promising therapeutic target and biomarker in BRCA1-associated breast cancers.

## ELF3 helps maintain the stability of DNA replication

We then investigated the mechanism by which ELF3 helps tolerate replication stress. ELF3 deficiency leads to more endogenous DNA damage in MCF10A cells (*Figure 5—figure supplement 1A*). In the BRCA1-deficient breast cancer cell line HCC1937, knockdown of ELF3 resulted in a significant increase in spontaneous γH2AX foci (*Figure 5A*), demonstrating that ELF3 deficiency leads to more endogenous DNA damage in BRCA1-deficient breast cancer cells. Moreover, with HU treatment and release for repair, ELF3 deficiency resulted in a significant increase in the number of γH2AX and 53BP1

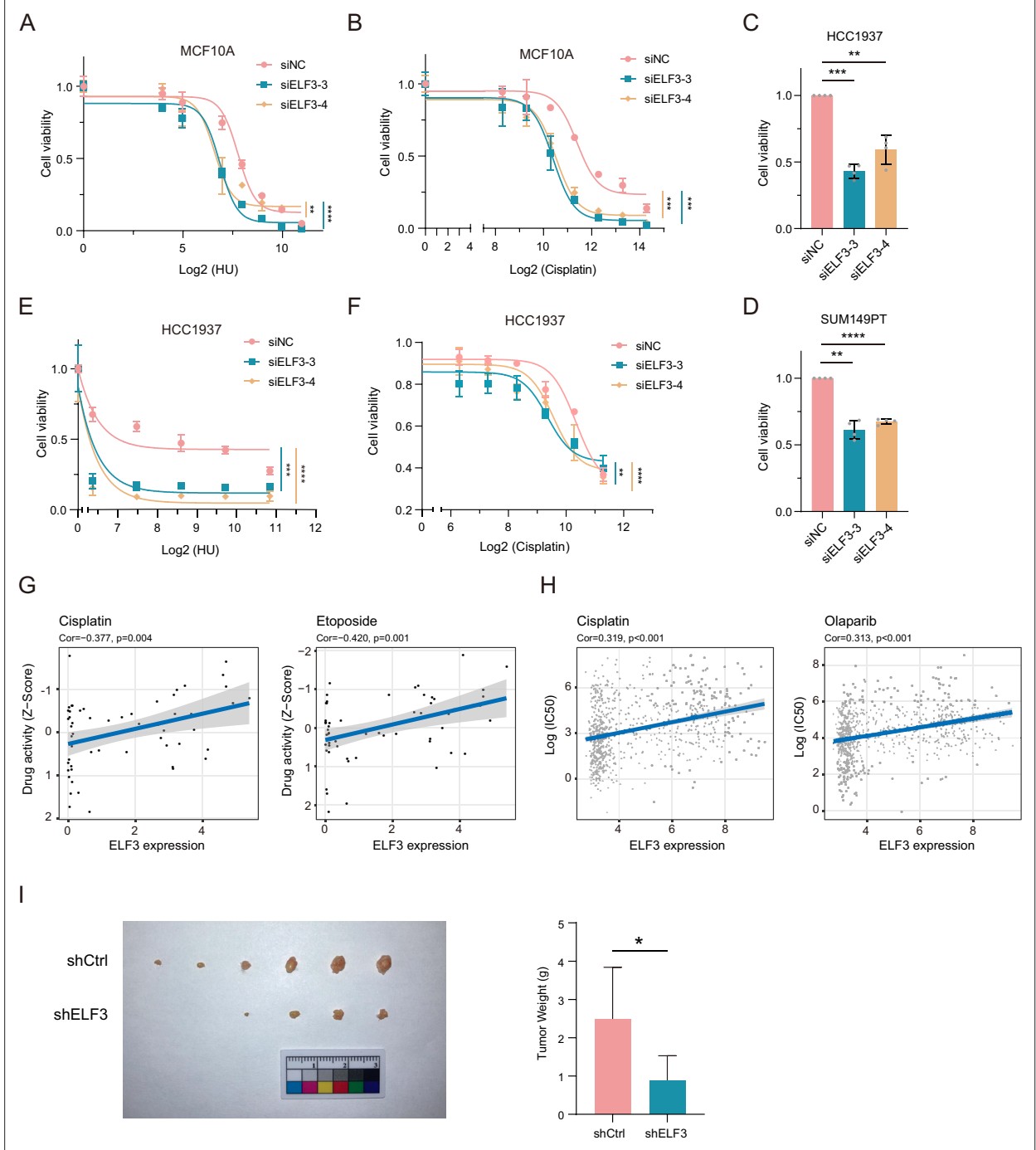

**Figure 4.** ELF3 helps suppress excessive genomic instability. (**A** and **B**) Cell viability curve of MCF10A cells transfected with siNC or siELF3 for 48 hr, treated with the indicated drugs, and measured by the CCK-8 assay (mean ± SD, n=3, two-way ANOVA, **$p<0.01$; ***$p<0.001$; ****$p<0.0001$). (**C** and **D**) Cell viability of HCC1937 (**C**) and SUM149PT (**D**) cells transfected with siNC or siELF3 for 5 days detected by the CCK-8 assay (mean ± SD, n=4, paired two-tailed Student's t-test, **$p<0.01$; ***$p<0.001$; ****$p<0.0001$). (**E** and **F**) Cell viability curve of HCC1937 cells transfected with siNC or siELF3 for 48 hr, treated with the indicated drugs, and measured by the CCK-8 assay (mean ± SD, n=3, two-way ANOVA, **$p<0.01$; ***$p<0.001$; ****$p<0.0001$). (**G** and **H**) Correlation of ELF3 expression levels and cell sensitivity of the indicated drugs in the CellMiner (**G**) and GDSC (**H**) databases. (**I**) Nude mouse tumors of SUM149PT cells infected with shCtrl or shELF3 lentivirus (mean ± SD, unpaired two-tailed Student's t-test, *$p<0.05$, shCtrl n=6, shELF3 n=5).

The online version of this article includes the following source data and figure supplement(s) for figure 4:

**Figure supplement 1.** ELF3 helps suppress excessive genomic instability.

**Figure supplement 1—source data 1.** Original western blots for *Figure 4—figure supplement 1A, B, C, E* indicating the relevant bands.

**Figure supplement 1—source data 2.** Original files for western blot analysis displayed in *Figure 4—figure supplement 1A, B, C, E*.

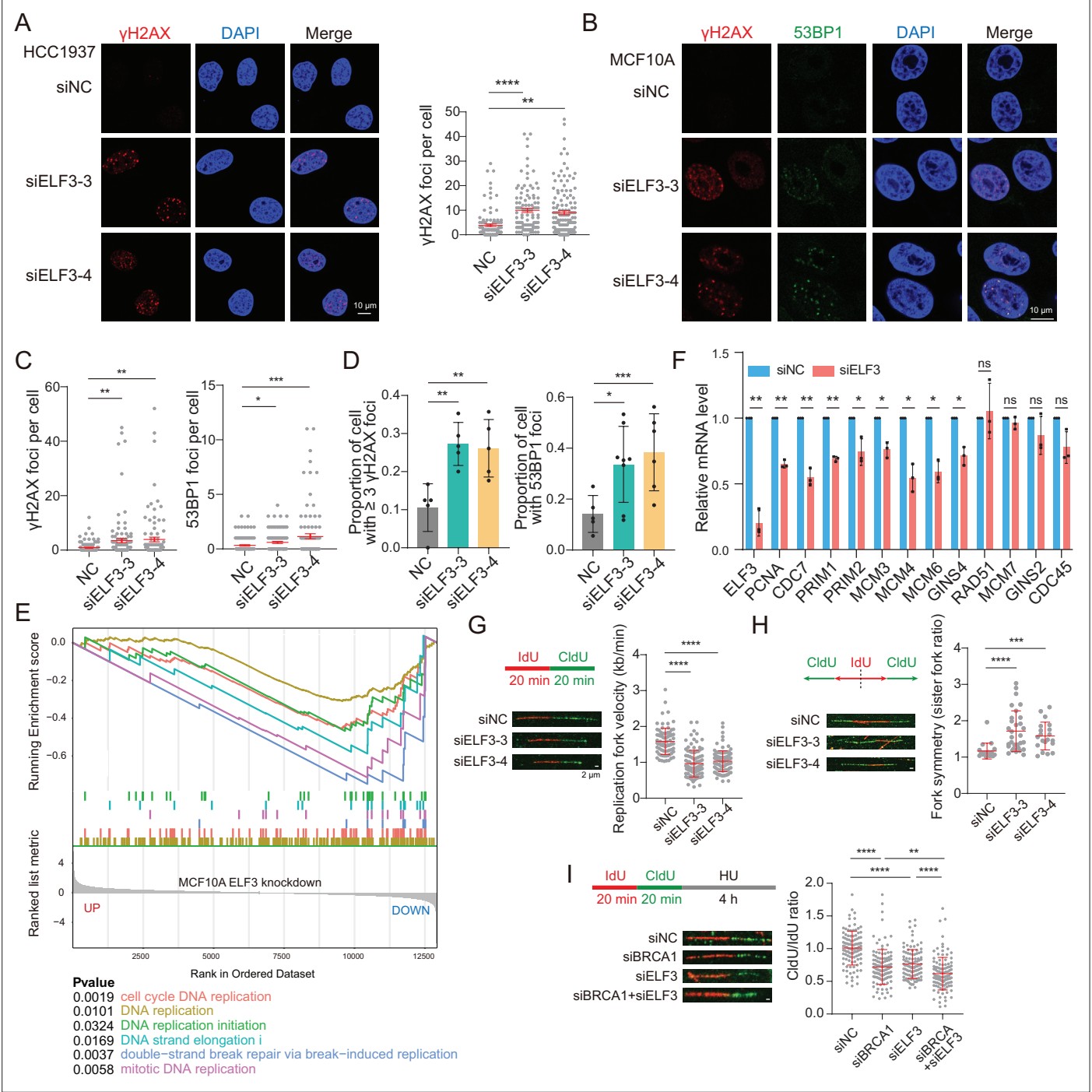

**Figure 5.** ELF3 helps maintain the stability of DNA replication. (**A**) Spontaneous γH2AX foci of HCC1937 cells transfected with siNC or siELF3 for 72 hr (mean ± SEM, n>100, unpaired two-tailed Student's t-test, **p<0.01; ****p<0.0001). (**B and C**) γH2AX and 53BP1 foci of MCF10A cells transfected with siNC or siELF3 for 48 hr and treated with 2 mM HU for 4 hr followed by release for 20 hr (mean ± SEM, n>100, unpaired two-tailed Student's t-test, *p<0.05; **p<0.01; ***p<0.001). (**D**) Proportions of cells with γH2AX (≥3) and 53BP1 foci from (**B and C**) (mean ± SD, unpaired two-tailed Student's t-test, *p<0.05; **p<0.01; ***p<0.001). (**E**) Enrichment plot of DNA replication pathways in MCF10A ELF3 knockdown RNA seq. (**F**) RT–qPCR results of MCF10A cells transfected with siNC or siELF3 (#4) for 72 hr (mean ± SD, n=3, paired two-tailed Student's t-test, *p<0.05; **p<0.01; ns, no significance). (**G**) Replication fork velocity indicated by DNA fiber assay of MCF10A cells transfected with the indicated siRNA for 72 hr (mean ± SD, n>80, unpaired two-tailed Student's t-test, ****p<0.0001). (**H**) Replication fork symmetry indicated by the DNA fiber assay of MCF10A cells transfected with the indicated siRNA for 72 hr (mean ± SD, Mann–Whitney test, ***p<0.001; ****p<0.0001). (**I**) Replication fork stability of MCF10A cells transfected with the indicated siRNA for 72 hr (mean ± SD, n>100, unpaired two-tailed Student's t-test, **p<0.01; ****p<0.0001).

*Figure 5 continued on next page*

*Figure 5 continued*

The online version of this article includes the following figure supplement(s) for figure 5:

**Figure supplement 1.** ELF3 helps maintain the stability of DNA replication.

foci per cell and the proportion of cells with damage foci (*Figure 5B–D*). Under replication stress conditions, γH2AX and 53BP1 foci represent DNA double-strand breaks formed by collapsed replication forks (*Pathania et al., 2011*). These data suggest that ELF3 deficiency results in a defective ability to repair stalled replication forks; thus, more DNA damage is generated. The upregulation of ELF3 expression during replication stress helps cells better manage stalled replication forks and prevents excessive DNA damage.

To further explore how ELF3 contributes to replication stress management, RNA-seq on ELF3-knockdown MCF10A cells was performed. Gene set enrichment analysis (GSEA) showed that multiple pathways associated with DNA replication were significantly downregulated in ELF3-deficient cells (*Figure 5E*), which was further verified by RT–qPCR (*Figure 5F*, *Figure 5—figure supplement 1B*). These data indicate that DNA replication pathways are affected by ELF3 deficiency.

A DNA fiber assay was performed to more closely investigate the effects of ELF3 deficiency on DNA replication. ELF3 deficiency significantly slowed down the DNA replication progression (*Figure 5G*), with significantly reduced symmetry of replication forks (*Figure 5H*), suggesting that more replication forks were stalled or unstable. Moreover, with HU treatment, ELF3 knockdown undermined the stability of stalled replication forks (*Figure 5I*). Notably, double knockdown of ELF3 and BRCA1 led to a further decrease in replication fork stability (*Figure 5I*). These data reveal that ELF3 expression upregulation when BRCA1 is deficient can help cells deal with replication stress, while simultaneous ELF3 deficiency leads to intolerable levels of genomic stability.

## ELF3 is a core transcription factor inducing LP gene expression under BRCA1 deficiency

Finally, we explored the possible reasons for the tissue specificity of BRCA1-associated tumors. It has been reported that BRCA1 deficiency disturbs the development and differentiation of mammary tissue and causes abnormal expansion of the LP population (*Lim et al., 2009*; *Hu et al., 2021*; *Liu et al., 2008*). We hypothesized that BRCA1 deficiency might affect the normal differentiation process of LPs by disturbing the transcriptional profile, thus leading to abnormal expansion and transformation of LPs. We analyzed the time-course RNA-seq data using a published signature gene set from LPs (*Lim et al., 2009*) and found that over time with BRCA1 deficiency, normal mammary cells gradually show a higher signature score of LP gene expression (*Figure 6A*). Similarly, using the sets of genes up- and downregulated in the LP population compared to other mammary cell populations obtained from the data of *Pellacani et al., 2016*, we found that genes upregulated in LPs also tended to be upregulated in BRCA1-deficient conditions, whereas genes downregulated in LPs tended to be downregulated (*Figure 6B*). These results suggest that BRCA1 may broadly affect the transcription profile of mammary cells, and therefore, when BRCA1 is deficient, LPs may tend to maintain a dedifferentiated state, leading to the abnormal expansion and tumorigenic properties of these cells in *BRCA1* mutant carriers.

Multiple ETS family transcription factors were shown to be involved in human mammary development and differentiation in previous reports. However, the function of ELF3 in LPs has not yet been elucidated. We used GSEA to explore the transcriptional profiles of BRCA1-deficient mammary cells, ELF3-overexpressing mammary cells, and human LPs from previous reports (*Pellacani et al., 2016*). Surprisingly, these three transcriptional profiles have significant correlations (*Figure 6C*), indicating that ELF3 is not only a core transcription node under BRCA1 deficiency conditions but also, more importantly, that ELF3 plays significant roles in the transcriptional regulation of human LPs. In addition, in our single-cell sequencing data, the expression of ELF3 in LPs is the highest among all mammary cell populations (*Figure 6D*, *Figure 6—figure supplement 1A*). Same results were obtained by analyzing the published data from *Pellacani et al., 2016*; *Figure 6—figure supplement 1B*. To investigate ELF3 functions in regulating LP genes transcription, we used SEA (simple enrichment analysis) of MEME to perform motif enrichment of ESE and ELF subfamilies transcription factors in promoters of LP genes, as these transcription factors are more closely related to ELF3 in the ETS family, and several,

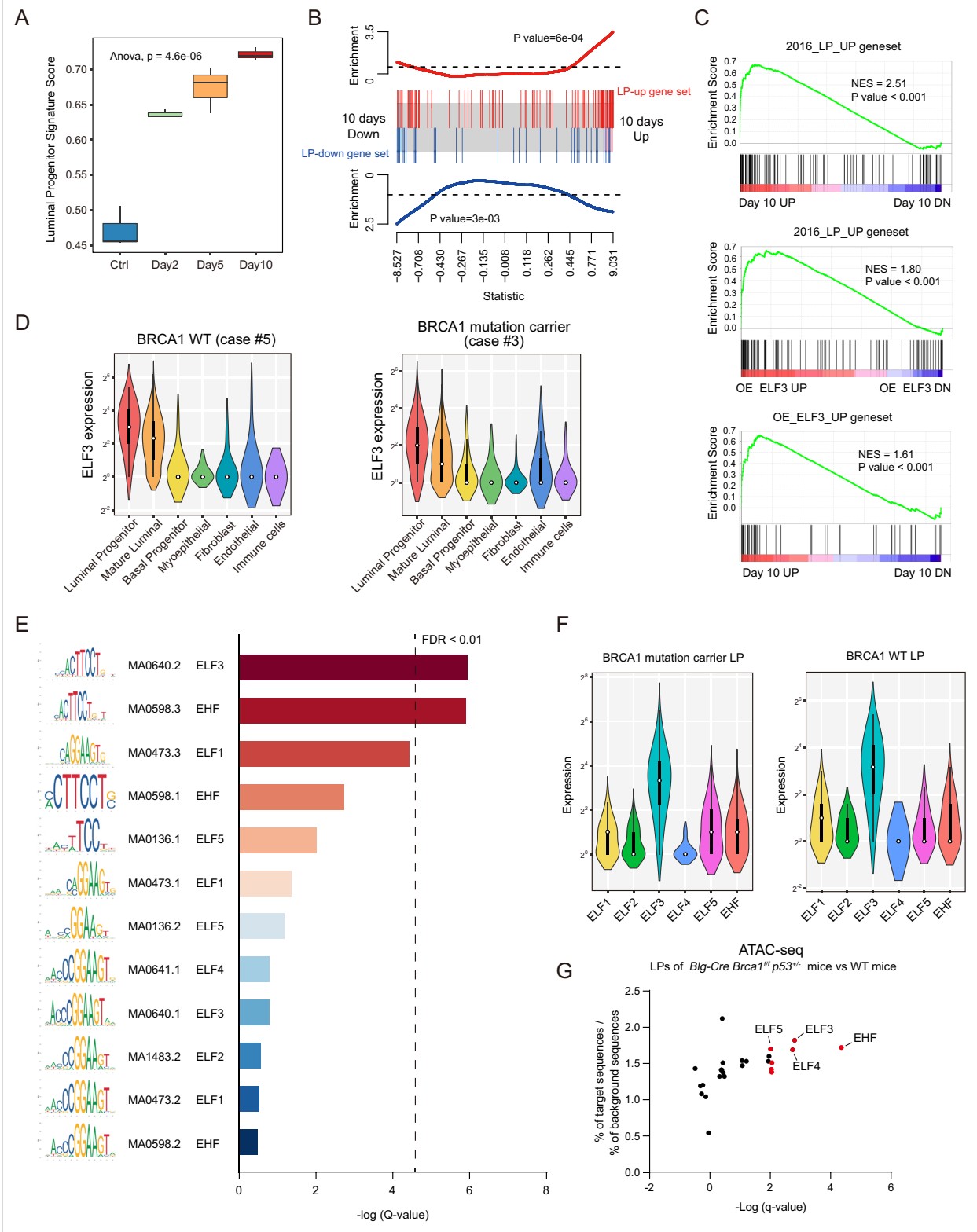

**Figure 6.** BRCA1 deficiency disturbs the differentiation of luminal progenitor cells (LPs). (**A**) LP signature score of samples of MCF10A-shBRCA1-Tet-on RNA-seq. The signature score was generated using data from ***Lim et al., 2009***. The bars (whiskers) in the boxplots represent the minimum and maximum values within the range of Q1-1.5*IQR and Q3+1.5*IQR, but do not include outliers. n=3 for each group(**B**) Barcode plot of LP gene set enrichment in Day 10 vs. Ctrl. LP gene sets were obtained from data from Pellacani et al. (***Shiah et al., 2015***). (**C**) Gene set enrichment analysis (GSEA) plots of the transcriptional profile of MCF10A cells with BRCA1 deficiency, MCF10A cells with ELF3 overexpression, and normal human LPs. The LP gene

*Figure 6 continued on next page*

*Figure 6 continued*

set was obtained from the data of Pellacani et al. (*Shiah et al., 2015*). (**D**) Violin plots displaying the distribution of ELF3 expression levels in distinct cell types in normal mammary tissue from non-carriers (left) and *BRCA1* mutation carriers (right). (**E**) ETS subfamily transcription factor motif enrichment results of LP gene promoters. (**F**) Expression levels of ETS subfamily transcription factors in LPs from *BRCA1* mutation carriers and non-carriers of the single-cell sequencing data. (**G**) ETS subfamily transcription factor enrichment of ATAC-Seq of LPs of BRCA1-deficient mice and wild-type (WT) mice from Bach et al. (*Sedic et al., 2015*).

The online version of this article includes the following figure supplement(s) for figure 6:

**Figure supplement 1.** Luminal progenitor cells (LPs) have a trait of higher replication stress, endowing them with the potential for transformation with BRCA1 deficiency.

such as ELF5 and EHF, have been reported to regulate LPs (*Pellacani et al., 2016*; *Oakes et al., 2008*; *Chakrabarti et al., 2012*). ELF3 was at the top of the list (*Figure 6E*), demonstrating that ELF3 has higher affinity for the promoters of LP genes. Moreover, our single-cell data revealed that ELF3 expression levels were the highest in LPs compared to other members of the ESE and ELF subfamilies (*Figure 6F*), including ELF5, which has been reported to play vital roles in LP differentiation. In addition, we analyzed the ATAC-seq data in Brca1-deficient mouse LPs from *Bach et al., 2021*. The ATAC-seq compared the chromatin accessibility of LPs in Brca1/p53-deficient mice with WT mice, and motif enrichment analysis of transcription factors was performed in regions with increased accessibility of Brca1-deficient mouse LPs. ELF3 was among the top genes (*Figure 6—figure supplement 1C*) and is significantly enriched among all ETS transcription factors (*Figure 6G*). These data further confirm our hypothesis that ELF3 plays an important role in the transcriptional regulation of LPs.

## Discussion

In this study, we revealed that LPs have the trait of higher replication stress, an important factor for their potential to transformation. The difference in LPs from other cell populations in normal mammary tissue has received considerable attention in the field. Research in mouse mammary tissue has revealed that the overall luminal epithelial cell population is enriched in mitotic, cell cycle, and DNA repair-related pathways under estrogen- and progestogen-stimulated conditions relative to the basal and stromal cell populations (*Shiah et al., 2015*). Subsequently, *Casey et al., 2018* found that LPs of mouse mammary tissue are enriched in DNA repair and cell division pathways. A recent preprint from the same group (*Kim et al., 2021*) continued the analysis of mouse tissue sequencing data and found that in LPs, DNA repair and replication pathways are significantly upregulated, indicating that LPs are more capable of DNA damage repair and thus more tolerant to BRCA1 deficiency. Furthermore, *Nolan et al., 2016* found that RANK+ LPs are highly proliferative and prone to DNA damage. These studies did not reveal the reason for LP's tendency to transform. In our study, we focus on the characteristics of LPs compared to other cell populations, and using sequencing data from normal human mammary tissues (*Lim et al., 2009*), we reveal that the DNA replication pathways and the cellular response to replication stress pathways are significantly upregulated in LPs compared to those in other cell populations. We argue that high levels of replication stress may be the major source of DNA damage in LPs, resulting in DDR gene upregulation in these cells, and the high proliferation feature of LP could be the main reason for the high replication stress.

Replication stress is a hallmark of cancer, fueling cancer evolution by increasing genomic instability (*Macheret and Halazonetis, 2015*). LPs have inherently higher replication stress levels, and when BRCA1 is deficient, replication stress is further elevated during the transformation from normal LPs to tumor cells, suggesting that replication stress is an important driver of BRCA1-deficient tumorigenesis and the suppression of replication stress could be a new approach for preventive intervention in *BRCA1* mutation carriers with the advantages of being less damaging than prophylactic resection. It was reported that haploinsufficiency in breast epithelial cell cultures from *BRCA1* heterozygotes could lead to premature failure through replication stress (*Sedic et al., 2015*). Moreover, we found that both replication stress and BRCA1 deficiency itself can induce high expression of ELF3, which also plays an important role in the tendency of LPs to transform with BRCA1 deficiency. First, ELF3 can help suppress excessive genomic instability, which is crucial for cell tolerance and survival with BRCA1 deficiency. We speculate that high ELF3 expression can help LPs maintain moderate levels of genomic instability that allow cells to proliferate and is sufficient to fuel cancer evolution, including

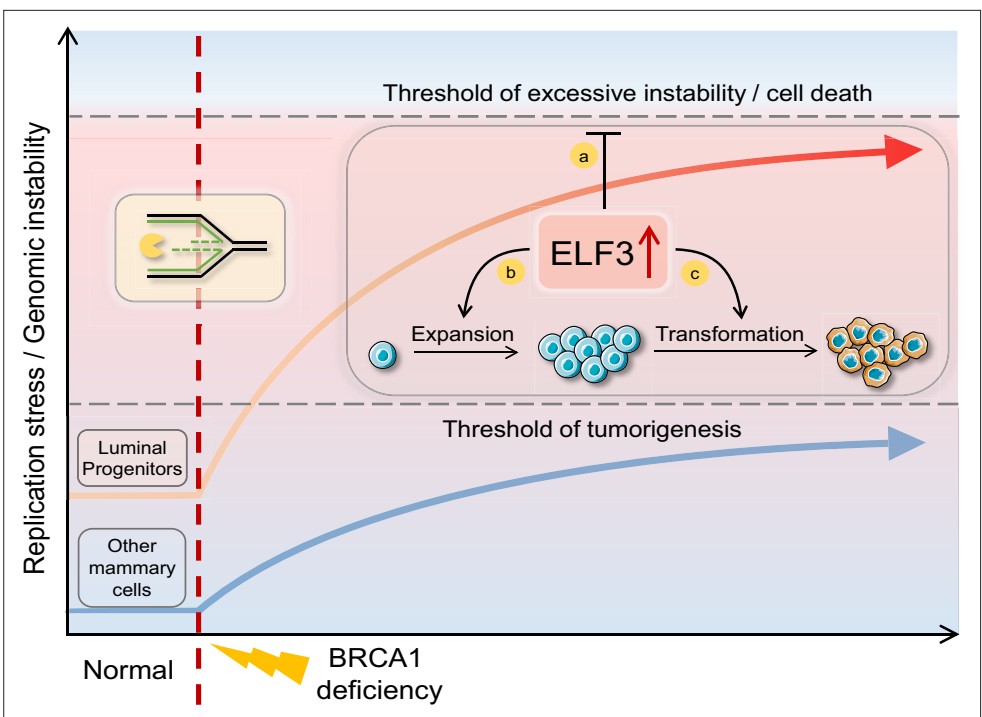

**Figure 7.** Replication stress-inducing ELF3 upregulation promotes BRCA1-deficient breast tumorigenesis. Luminal progenitor cells (LPs) have higher replication stress levels under normal conditions (left) and thus are more likely to exceed the tumorigenesis threshold under BRCA1 deficiency (right). Replication stress and BRCA1 deficiency result in further ELF3 upregulation (top right). ELF3 can suppress excessive replication stress and help LPs maintain moderate levels of genomic instability, which is tolerable for cells to proliferate and enough to fuel cancer evolution (a). In non-LP mammary cells, which exhibit inherently low replicative stress, the ELF3 upregulation may enable cells to endure the increased replicative stress caused by BRCA1 deficiency without leading to malignancy. However, in LP cells, which naturally experience higher levels of replicative stress, this ELF3-mediated mechanism may increase susceptibility to cancerous transformation. In addition, ELF3 upregulation can boost LP gene transcription, leading to LP dedifferentiation and expansion (b). Finally, as an EMT promoter in the mammary tissue, ELF3 can accelerate the transformation process from LPs to malignant cells (c).

subsequent events, such as *p53* mutation and *BRCA1* loss of heterozygosity (LOH). Second, ELF3 upregulation can boost LP gene transcription, leading to LP dedifferentiation and expansion. Finally, as an EMT promoter in mammary tissue, ELF3 can accelerate the transformation of LPs to malignant cells. Therefore, high levels of replication stress and ELF3 expression are important factors resulting in the high tendency for LPs to transform (*Figure 7*).

A key feature of BRCA1-associated tumors is the tissue specificity, the mechanism of which has long been a research focus. The transcriptional regulation functions of BRCA1 are critical for the development and differentiation of normal mammary tissue. LPs with abnormal expansion have been observed in *BRCA1* mutation carriers. Several important differentiation pathways, such as WNT, Notch, and Hedgehog, were downregulated in these abnormal LPs (*Hu et al., 2021*), suggesting that BRCA1 deficiency may lead to a differentiation blockade in LPs and, therefore, abnormal expansion. Our RNA-seq data revealed that the transcriptional profile of BRCA1-deficient MCF10A cells has a striking similarity to that of LPs in normal human mammary tissue, and ELF3 might be a core mediating transcription factor. This finding suggests that BRCA1 has a broad regulatory effect on gene expression in normal mammary cells and has specific functions in LPs; thus, its deficiency causes LPs to maintain a dedifferentiated state, leading to abnormal expansion of this cell population, and ELF3 may be a core transcription factor inducing these changes. In addition, our study reveals another layer of the mechanism underlying the tissue specificity of BRCA1-associated tumors—the capability of BRCA1 to regulate ELF3 expression. ELF3 is an epithelial-specific transcription factor, but the functions of ELF3 in different epithelial tissues vary. For example, ELF3 promotes tumor development in breast, prostate, and lung cancers (*Chang et al., 1997*; *Longoni et al., 2013*; *Enfield et al., 2019*),

whereas it suppresses tumorigenesis in bladder and oral squamous cancers (*Iwai et al., 2008*; *The Cancer Genome Atlas Research Network, 2014*). In addition, ELF3 itself plays an important role in the development and differentiation of normal mammary tissue; thus, dysregulation of ELF3 may disturb these processes. Therefore, we suggest that replication stress caused by BRCA1 deficiency leads to upregulation of ELF3 expression in LPs, thus promoting the tissue-specific oncogenic functions of ELF3 and breast tumorigenesis.

This study also has some limitations. First, we used the MCF10A cell line, which is widely used for investigations into normal human mammary epithelial cells with the advantage of being easy to manipulate and modify, and compared to patient-derived cells, unknown genetic backgrounds in addition to *BRCA1* mutations are avoided. However, the transcriptional profile and epigenetic characteristics of this cell line are distinct from those of normal human mammary cell populations and LPs (*Pellacani et al., 2016*); thus, there are certain gaps between the cell model and real LP cells. Although our data show that MCF10A cells can reflect the actual functions of BRCA1 in terms of maintaining genomic stability (*Figure 1B and C*) and our RNA-seq data demonstrate enrichment in the known transcriptional target genes of BRCA1, the effect of BRCA1 defects and replication stress in LPs needs to be further verified in primary human mammary cell populations. An analysis of TNBC as well as basal-like tumor data needs to be further confirmed in BRCA1-deficient breast cancer. Second, the higher replication stress levels in LPs may lead to altered expression or modifications of molecules other than ELF3, affecting the development and progression of BRCA1-associated breast cancer, which also needs to be further explored. In addition, there are likely other DEGs in our BRCA1-deficient RNA-seq that play important roles in the progression of BRCA1-associated breast cancer, and these genes also have potential research value.

Taken together, we revealed that LPs have a trait of higher replication stress, endowing them with the potential for transformation with BRCA1 deficiency. Moreover, the levels of replication stress gradually increase during tumorigenesis, indicating that replication stress is a driving factor of BRCA1-associated malignant transformation. Mechanistically, ELF3 is significantly upregulated upon replication stress and BRCA1 deficiency, which can help suppress excessive genomic instability by stabilizing the DNA replication process, empowering LPs to tolerate and to transform with BRCA1 deficiency. In cells with inherently low replicative stress, such as other non-LP mammary cells, the ELF3-associated mechanism might help cells endure the high replicative stress caused by BRCA1 deficiency without leading to cancerous changes. However, in LP cells, which naturally experience higher replicative stress, this ELF3-related mechanism may make them more susceptible to transformation into cancer cells. Notably, BRCA1 deficiency causes a tendency for cells to display characteristics of the LP transcriptional profile, and ELF3 is a core transcription factor mediating this process. Our study reveals why BRCA1 deficiency is prone to result in tumorigenesis in LPs and elucidates the key role of replication stress and ELF3 during this process and suggests promising targets for BRCA1-associated breast cancers.

## Materials and methods

**Key resources table**

| Reagent type (species) or resource | Designation | Source or reference | Identifiers | Additional information |
|---|---|---|---|---|
| Gene (*Homo sapiens*) | ELF3 | HGNC | HGNC:3318 | |
| Strain, strain background (include species and sex here) | BALB/c nude mice Female | Charles River | 401 | |
| Cell line (*Homo sapiens*) | MCF10A | Zomanbio | ZKC1102 | |
| Cell line (*Homo sapiens*) | SUM149PT | BMCR | 3101HUMSCSP5090 | |
| Cell line (*Homo sapiens*) | HCC1937 | BMCR | 1101HUM-PUMC000471 | |
| Antibody | anti-BRCA1 (Mouse monoclonal) | Merck-Millipore | OP92 | WB (1:1000) |

*Continued on next page*

*Continued*

| Reagent type (species) or resource | Designation | Source or reference | Identifiers | Additional information |
|---|---|---|---|---|
| Antibody | anti-ELF3 (Rabbit monoclonal) | Abcam | Ab133621 | WB (1:2000) |
| Recombinant DNA reagent | Tet-pLKO-hygro-shBRCA1 | This paper | | shBRCA1 version of Tet-pLKO-hygro |
| Recombinant DNA reagent | pCDH-CMV-MCS-EF1-puro-ELF3 | This paper | | ELF3 version of pCDH-CMV-MCS-EF1-puro |
| Recombinant DNA reagent | pLKO.1-shBRCA1 | This paper | | shBRCA1 version of pLKO.1 |
| Recombinant DNA reagent | pLKO.1-shELF3 | This paper | | shELF3 version of pLKO.1 |
| Sequence-based reagent | ELF3 siRNA#3 sense | This paper | siRNA sequences | GAAGUGACGUGGACCUGGATT |
| Sequence-based reagent | ELF3 siRNA#3 anti-sense | This paper | siRNA sequences | UCCAGGUCCACGUCACUUCCA |
| Sequence-based reagent | ELF3 siRNA#4 sense | This paper | siRNA sequences | GCCGAUGACUUGGUACUGATT |
| Sequence-based reagent | ELF3 siRNA#4 anti-sense | This paper | siRNA sequences | UCAGUACCAAGUCAUCGGCCC |
| Sequence-based reagent | BRCA1 siRNA sense | This paper | siRNA sequences | CAGCUACCCUUCCAUCAUATT |
| Sequence-based reagent | BRCA1 siRNA anti-sense | This paper | siRNA sequences | UAUGAUGGAAGGGUAGCUGTT |
| Chemical compound, drug | Doxycycline hydrochloride | HARVEYBIO | D31646 | |
| Software, algorithm | Mfuzz | PMID:18084642 | Mfuzz 2.46.0 | |

## Cell culture

MCF10A cells were cultured in DMEM/F12 supplemented with 5% horse serum, 1% penicillin/streptomycin, 20 ng/mL EGF, 10 μg/mL insulin, 0.5 μg /mL hydrocortisone, and 100 ng/mL cholera toxin. For MCF10A-shBRCA1-Tet-on cells, except for the replacement of horse serum with 8% Tet-on system-approved certified fetal bovine serum, all other components remain the same. HEK293T and SUM149PT cells were cultured in DMEM supplemented with 10% fetal bovine serum and 1% penicillin/streptomycin. HCC1937 cells were cultured in RPMI 1640 supplemented with 10% fetal bovine serum and 1% penicillin/streptomycin. All cell lines used in this study were authenticated by short tandem repeat (STR) profiling and tested negative for mycoplasma contamination using PCR-based detection assays prior to experimental use. None of the cell lines used are listed among the commonly misidentified cell lines according to the International Cell Line Authentication Committee (ICLAC).

## Reagents and plasmids

Reagents (antibodies and chemicals) used in this study were listed in *Supplementary file 3*. For the construction of the Tet-on-shBRCA1-hygro plasmid, shBRCA1 sequence was subcloned into the Tet-on-pLKO-hygro vector (SinaSun). For ELF3 knockdown by lentivirus in SUM149PT, shELF3, and shCtrl sequence was subcloned into the pLKO.1-puro vector. The shRNA sequences used in this study are listed in *Supplementary file 3*. For ELF3 overexpression by lentivirus in MCF10A, human ELF3 was subcloned into the pCDH-CMV-MCS-EF1-puro vector, which was a kind gift from Yin's lab. Myc-tagged human E2F6 was subcloned into the pDEST-N-Myc vector. PCR primers used in this study were listed in *Supplementary file 3*.

## Transfection of siRNA and infection of shRNA

All the siRNAs were from GenePharma. For siRNA transfection, cells were seeded in a 6-well plate or 35 mm$^2$ dish the day before transfection. 100 μL OPTI-MEM, 40 pmol siRNA, and 3 μL Lipofectamine RNAiMAX reagent were mixed and added to the cells. After 48–72 hr, cells were collected for subsequent experiments. The siRNA sequences used are listed in *Supplementary file 3*. For shRNA infection, lentivirus was generated by co-transfection of the lentiviral vector (Tet-pLKO.1-shBRCA1, pLKO.1-shCtrl, and pLKO.1-shELF3) with envelope plasmids pMD2.G and psPAX2 in HEK293T cells. Lentivirus was collected after 48 hr. Cells were infected with lentivirus with polybrene. After 24 hr, cells were re-infected and harvested at 72 hr for subsequent experiments.

## Quantitative PCR (RT-qPCR)

Total RNA was extracted using Trizol reagent (Invitrogen) according to the manufacturer's guidelines. cDNA was synthesized by reverse transcription PCR using Hifair III 1st Strand cDNA Synthesis SuperMix kit (Yeasen). qPCR was performed using SYBR Green Mix (Yeasen) on ABI StepOnePlus qPCR instrument (Thermo Fisher). Each group was performed in triplicate. The relative expression mRNA levels were calculated by the relative quantitative method ($\Delta\Delta Ct$). The primers used are listed in **Supplementary file 3**.

## Immunofluorescence

Cells were seeded on coverslips and treated as indicated. Cells were then fixed in cold 4% paraformaldehyde for 15 min and permeabilized in 0.25% Triton X-100 solution for 5 min at room temperature. After blocking with 2% BSA for 10 min, the primary antibodies were added onto the coverslips and then incubated for 1 hr at room temperature, followed by secondary antibody incubation for 30 min. Finally, the coverslips were stained with DAPI for 5 min, and images were acquired using the ZEN software.

## Immunohistochemistry

All tissue microarray chips were purchased from Shanghai Outdo Biotech Co. Ltd. IHC was performed as described previously, except that the following antibodies were used. The anti-ELF3 rabbit antibody were purchased from SIGMA. After incubation with the primary antibody, the HRP-conjugated rabbit secondary antibody (ZSGB-Bio) was applied and IHC slides were observed with an Olympus BX51 microscope and anOlympus DP73 CCD photographic system.

## Colony formation assay

MCF10A-shBRCA1-Tet-on cells were seeded in 6 mm$^2$ plates. 1 µg/mL DOX was added for the DOX group during the whole process of cell growth to knock down BRCA1 expression. After 14 days, the cells were washed with phosphate-buffered saline (PBS) and stained with 0.1% Coomassie brilliant blue in 10% ethanol for 30 min at room temperature. The stained dishes were washed with water, and the colonies were counted.

## DNA fiber

The DNA fiber assay was performed as described previously with slight modifications (**Nieminuszczy et al., 2016**). Briefly, cells transfected with siRNA were labeled with 25 µM IdU (Yuanye Bio-Technology) for 20 min and subsequently 200 µM CldU (Abcam) for 20 min. Then, cells were treated with 4 mM HU for 4 hr for replication fork stability assay or collected for replication fork speed and symmetry assay. Cells were resuspended in lysis buffer (200 mM Tris-HCl, pH 7.5, 50 mM EDTA, and 0.5% SDS) to extract DNA. Cell lysates were then dripped onto a glass slide, and DNA fibers were stretched along the slide after 3 min incubation. The slides were air-dried and fixed in a 3:1 methanol/acetic acid solution for 10 min, followed by a 2.5 M HCl treatment for 1 hr, then blocked in 3% BSA PBST buffer for 20 min. DNA fibers were incubated with anti-BrdU antibody (Abcam) binding CldU and anti-BrdU antibody (BD) binding IdU at 37°C for 1.5 hr. After washing with PBST, the slides were incubated with Alexa Fluor 488- and Alexa Fluor 594-conjugated secondary antibody (Invitrogen) for 45 min. Finally, the slides were mounted in mounting reagent (Solarbio), and DNA fiber images were acquired using ZEN software.

## Comet assay

The comet assay was performed as described previously with slight modifications (**Rojas et al., 1999**). Briefly, cells were collected by trypsin digestion and resuspended in PBS. 0.5% normal melting agarose (NMA) was added to glass slides and quickly covered with coverslips. The slides were placed at 4°C for 10 min. The coverslips were removed carefully. The cell suspensions were mixed gently with an equal volume of 1% low melting agarose (LMA) and quickly added onto the solidified NMA, quickly covered with coverslips. The slides were placed at 4°C for 10 min. After removing the coverslips, the slides were incubated overnight in the fresh cell lysis buffer at 4°C overnight. Then the slides were rinsed with ddH$_2$O and electrophoresis at 4°C for 30 min at 20 V in neutral electrophoresis buffer. Then the slides were stained with 5 µg/ml of propidium iodide for 10 min. The comet was observed

using a fluorescence microscope (Zeiss AxioCam 503 color). The pictures were analyzed using CASP software (http://casplab.com/).

## Cell survival (CCK-8) assay

Cell survival assay was performed using CCK-8 kits (Meilunbio, MA0218-3) according to the manufacturer's protocols. Cells were seeded in 96-well plates. After 12 hr, medium containing indicated concentration of drugs was changed and cells were incubated for 3–5 days. DMEM containing 0.1% DMSO was used as the control group. CCK-8 solution was diluted in 10% concentration and added to wells, and the cells were incubated for 45 min-1.5 hr. DMEM containing 10% CCK-8 was used as the blank group. The absorbance at 450 nm was detected by a microplate reader. Cell survival was calculated according to the formula: (ODdrug – ODblank)/(ODcontrol – ODblank).

## RNA sequencing

Total RNA of cells was extracted with Trizol reagent. RNA sequencing was performed by Novogene (China). The resulting data was mapped to human reference genome (hg38) and gene counts were calculated for each sample using STAR (v2.6.1) (*Dobin et al., 2013*) with the parameter '--quantMode GeneCounts.' To identify differentially expressed genes (DEGs), genes with low expression levels (average counts per million reads <0.05) were filtered out. DESeq2 was employed to detect DEGs with |log2 fold change|>2 and FDR <0.001. PCA analysis was performed using the 'prcomp' function of stats (v3.6.1) R package.

## Analysis of TCGA breast cancer and METABRIC data

TCGA (https://www.cancer.gov/tcga) breast cancer and METABRIC (*Curtis et al., 2012*) data were downloaded using the cgdsr R package. For TCGA data, RSEM (batch normalized from Illumina HiSeq_RNASeqV2) was used as the expression levels of the genes. For METABRIC data, mRNA expression log intensity levels (Illumina Human v3 microarray) were used as the expression levels of the genes. BRCA1-associated cancers were identified as cancers with *BRCA1* mutations, heterozygous loss or homozygous deletion. The Pearson correlation coefficient between ELF3 expression and BRCA1 expression was calculated. Subtypes and IntClust types of breast cancers were identified according to the clinical data of TCGA and METABRIC databases.

## Chip-seq analysis of ENCODE databases

The Chip-seq data were downloaded from the ENCODE database (https://www.encodeproject.org/) (*Consortium, 2012*; *Davis et al., 2018*) with the following identifiers: ENCFF858GLM, ENCFF692OYJ, ENCFF384CPN, ENCFF342GNN, ENCFF352QVM, and ENCFF437NQS. Images were generated using IGV (v2.7.2).

## Drug sensitivity analysis of Cell Miner and GDSC data

Cell line drug sensitivity data and mRNA expression data were downloaded from the CellMiner (*Shankavaram et al., 2009*; *Reinhold et al., 2012*) and GDSC (*Yang et al., 2013*) databases. The Pearson correlation coefficient between ELF3 expression and drug sensitivity was calculated for the indicated drugs.

## Replication stress score and luminal progenitor score analysis

For replication stress score calculation, human mammary subgroups gene expression matrixes were obtained from the datasets of *Lim et al., 2009* (GSE16997). Replication stress-associated pathways were selected referring to *Dreyer et al., 2021*, and gene lists were downloaded from the GSEA website (http://www.gsea-msigdb.org). For each pathway, replication stress scores were calculated according to the formula below, where $e_g$ is the expression of the gene in different mammary subgroups, and is $n_g$ the number of genes in this pathway.

$$S = \sum_g e_g/n_g$$

For luminal progenitor score calculation, the LP signature gene set with average log-fold change was obtained from data of *Lim et al., 2009*, and log CPM matrix of MCF10A-shBRCA1-Tet-on DOX 10 days vs Ctrl RNA-seq data was used. For each sample, LP scores were calculated according to the formula below, which refers to *Lim et al., 2009*, where $x_g$ is the log-fold-change for the gene from LP gene set, and $y_g$ is the log cpm for the same gene in BRCA1 knockdown RNA-seq.

$$S = \sum_g x_g y_g / \sum_g |x_g|$$

## GSEA analysis and barcode plot

GSEA analysis was performed using GSEA software (v4.1.0) (*Mootha et al., 2003*; *Subramanian et al., 2005*) and clusterProfiler (v3.14.3) (*Yu et al., 2012*). For the customized gene set enrichment analysis of the transcriptional profiles of BRCA1 KD, ELF3 OE and LP, BRCA1 KD data were from MCF10A-shBRCA1-Tet-on DOX 10 days vs Ctrl RNA-Seq data and ELF3 OE data were from MCF10A OE ELF3 vs Ctrl RNA-seq data. The LP gene set was obtained from the 2016 cell reports, and OE ELF3 gene sets were generated using MCF10A OE ELF3 vs Ctrl RNA-seq data, where the log-fold-change cutoff was 1.3 and p-value cutoff was 0.01. For the barcode plot, LP gene sets were obtained from data of *Pellacani et al., 2016*, and log-fold change of all genes from MCF10A-shBRCA1-Tet-on DOX 10 days vs Ctrl RNA-seq data was used for the enrichment analysis. P-value was calculated using the 'roast' function from limma (v3.42.2) (*Ritchie et al., 2015*).

## Motif enrichment analysis

Motif matrices of transcription factors were downloaded from the JASPAR database (*Castro-Mondragon et al., 2022*). The LP gene list was obtained according to the data of *Pellacani et al., 2016* and LP genes are identified to be significantly upregulated in LPs compared with all three other subgroups (BC, LC, and SC). Promoter sequences of LP genes were extracted from the genome from 3000 bp upstream to 200 bp downstream of the TSS sites. Motif enrichment analysis was performed using SEA (Simple Enrichment Analysis) of the MEME Suite (*Bailey et al., 2015*).

## Single-cell sequencing data analysis

Single-cell RNA-seq (scRNA-seq) expression profiles of breast cancer tissue and normal mammary tissue from four *BRCA1* mutation carriers, as well as normal mammary tissue from three non-carriers were generated using the 10x Genomics Chromium platform and NovaSeq 6000 platform. The detailed information of sample processing, scRNA-seq data processing, and quality control were described in our previous study (*Hu et al., 2021*). Briefly, scRNA-seq data analyses was completed utilizing the 10x Genomics Cell Ranger software suite. Principal component analysis (PCA) (twenty-five principal components) and k-means algorithm (k was set as 9.0) were used to cluster cells. We used a series of cell-specific markers to identify the cell type of each cluster, as described in our previous study. Among these, a luminal progenitor cluster was identified based on the expression of KRT8/18+, CD24+, GABRP+, KIT+, and ALDH1A3+. A triple-negative tumor cell population was identified based on tumor sample-specific cluster, ER/PR/HER2 expression coincident with the clinical records and PAM50 subtypes analysis. Replication stress score of each cell was calculated based on the sum expression level of genes involved in replication stress-related pathways referring to *Dreyer et al., 2021*. In this study, we selected luminal lineage cells and tumor cells from a BRCA1-mutated triple-negative breast cancer patient to construct tumorigenesis trajectories with the R package Monocle 2 with standard settings.

## Animal experiments

$5 \times 10^6$ SUM149PT tumor cells were subcutaneously injected into the flanks of female BALB/c nude mice. Tumor growth was measured using a caliper every three days, and the tumor volume was calculated using the formula: tumor volume = length × width$^2$/2.

## Statistics analysis

Statistical analysis was performed using GraphPad Prism (GraphPad Software). The tests used included the paired and unpaired two-tailed Student's t test, the Mann-Whitney U test, one-way and two-way ANOVA, and Pearson's correlation statistical test. A p-value of less than 0.05 was considered statistically significant. Details can be found in figure legends.

## Study approval

The normal human breast cancer tissue and breast cancer samples used for single-cell RNA-seq in this study were collected from Peking University Cancer Hospital. This study was carried out in accordance with the ethical principles of the Declaration of Helsinki and was approved by the Research and Ethics Committee of Peking University Cancer Hospital. Written informed consent was obtained from all participants prior to participation. Animal experiments in this study were performed in accordance with the Guidelines of Peking University Animal Care and Use Committee.

## Acknowledgements

We thank Dr. Jianming Zeng (University of Macau), Xiaojie Sun, and all the members of the bioinformatics team, biotrainees, for generously sharing their experience and codes. We thank Ence Yang, Changyu Tao, and Zelin Liu for the kind sharing of computing resources and guiding on the RNA-seq data analytical methodology. We thank Lim et al., Pellacani et al., and Bach et al. for their outstanding contribution to BRCA1-associated breast tumorigenesis research and mammary cell populations sequencing data. We thank the ENCODE Consortium and the ENCODE production laboratories for generating the ChIP-seq datasets.

## Additional information

### Funding

| Funder | Grant reference number | Author |
|---|---|---|
| National Science Fund for Distinguished Young Scholars | 82125031 | Jiadong Wang |
| National Natural Science Foundation of China | 82230089 | Jiadong Wang |
| National Key Research and Development Program of China | 2022YFA1302800 | Jiadong Wang |
| National Natural Science Foundation of China | 82102997 | Xiao Albert Zhou |
| Postdoctoral Research Foundation of China | 2024M750126 | Xiao Albert Zhou |
| Postdoctoral Research Foundation of China | GZB20240037 | Xiao Albert Zhou |

The funders had no role in study design, data collection and interpretation, or the decision to submit the work for publication.

### Author contributions

Jiadong Zhou, Conceptualization, Data curation, Formal analysis, Validation, Investigation, Visualization, Methodology, Writing – original draft, Project administration; Xiao Albert Zhou, Conceptualization, Data curation, Formal analysis, Supervision, Funding acquisition, Validation, Investigation, Visualization, Methodology, Writing – original draft, Project administration, Writing – review and editing; Li Hu, Resources, Data curation, Formal analysis, Investigation; Yujie Ma, Data curation, Formal analysis, Investigation; Jun Zhan, Investigation, Methodology; Zhanzhan Xu, Formal analysis, Investigation,

Methodology; Mei Zhou, Qinjian Shen, Data curation, Methodology; Zhaofei Liu, Writing – review and editing; Shaohua Ma, Yuntao Xie, Resources, Investigation, Methodology, Writing – review and editing; Jiadong Wang, Conceptualization, Supervision, Project administration, Writing – review and editing

**Author ORCIDs**
Xiao Albert Zhou ⓘ https://orcid.org/0000-0002-3726-4523
Jiadong Wang ⓘ https://orcid.org/0000-0002-9431-2413

**Ethics**

The normal human breast cancer tissue and breast cancer samples used for single-cell RNA-seq in this study were collected from Peking University Cancer Hospital. This study was carried out in accordance with the ethical principles of the Declaration of Helsinki and was approved by the Research and Ethics Committee of Peking University Cancer Hospital. Written informed consent was obtained from all participants prior to participation.

Animal experiments in this study were performed in strict accordance with the Guidelines of the Peking University Animal Care and Use Committee. The protocol was reviewed and approved by the Laboratory Animal Welfare and Ethics Committee of Peking University Biomedical Ethics Committee (Approval Number: LA2021268). All procedures involving animals complied with institutional guidelines for the humane treatment of laboratory animals.

Reviewer #1 (Public review): https://doi.org/10.7554/eLife.89573.3.sa1
Author response https://doi.org/10.7554/eLife.89573.3.sa2

---

# Additional files

**Supplementary files**
MDAR checklist

Supplementary file 1. Expression matrix of genes for the replication stress signature (case 7).

Supplementary file 2. Detailed information on the BRCA1 germline mutation and clinicopathological characteristics of seven patients.

Supplementary file 3. Key reagents, oligonucleotide sequences, and primers used in this study.

**Data availability**

All the data supporting the findings of this study are available within the article and supplementary information file. The raw sequence data reported in this paper have been deposited in the Genome Sequence Archive (GSA-Human) of the National Genomics Data Center (accession No. HRA012853), and are publicly accessible at https://ngdc.cncb.ac.cn/gsa-human.

The following dataset was generated:

| Author(s) | Year | Dataset title | Dataset URL | Database and Identifier |
|---|---|---|---|---|
| Wang J | 2025 | Time-resolved RNA-seq of cells with BRCA1 deficiency | https://ngdc.cncb.ac.cn/gsa-human/browse/HRA012853 | HRA012853, GSA |

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
