## [Editor Report · eLife Assessment]

In this **fundamental** study, the authors describe ELF3 as a candidate driver of luminal progenitor transformation, such that its up-regulation during replicative stress conditions and in BRCA1 deficient cells may permit cell proliferation by suppressing genome instability. While the work is certainly of interest, the supporting data remain **incomplete** as luminal progenitor cells could not be isolated, which would be needed in order to definitively determine whether ELF3 is a driver of transformation in these cells. Overall this paper may offer insight into mechanisms by which BRCA1 deficiency fuels breast tumorigenesis.

---

## [Referee Report · Reviewer #1 (Public review)]

The authors set out to define the molecular basis for LP as the origin of BRCA1-deficient breast cancers. They showed that LPs have the highest level of replicative stress, and hypothesise that this may account for their tendency to transform. They went on to identify ELF3 as a candidate driver of LP transformation and showed that ELF3 expression is up-regulated in response to replicative stress as well as BRCA1 deficiency. They went on to show that ELF3 inactivation led to a higher level of DNA damage, which may result from compromised replicative stress responses.

While the manuscript supports the interesting idea wherein ELF3 may fuel LP cell transformation, it remains obscure how ELF3 promotes cell tolerance to DNA damage. Interestingly the authors proposed that ELF3 suppresses excessive genomic instability, but in my opinion, I do not see any evidence that supports this claim. In fact, one might think that genomic instability is key to cell transformation.

Comments on revisions:

The authors have addressed most of my concerns.

This being said, the one major criticism raised by both Reviewers is the lack of evidence to support ELF3 as a driver of transformation of and in LP cells. The authors appear to have invested much resource and time but were not successful in isolating LP cells for experimentations. I would therefore suggest that the authors tone down their claims throughout the manuscript.

---

## [Author Response]

The following is the authors’ response to the original reviews.

**Public Reviews:**

**Reviewer #1 (Public Review):**
The authors set out to define the molecular basis for LP as the origin of BRCA1deficient breast cancers. They showed that LPs have the highest level of replicative stress, and hypothesise that this may account for their tendency to transform. They went on to identify ELF3 as a candidate driver of LP transformation and showed that ELF3 expression is up-regulated in response to replicative stress as well as BRCA1 deficiency. They went on to show that ELF3 inactivation led to a higher level of DNA damage, which may result from compromised replicative stress responses.While the manuscript supports the interesting idea wherein ELF3 may fuel LP cell transformation, it remains obscure how ELF3 promotes cell tolerance to DNA damage. Interestingly the authors proposed that ELF3 suppresses excessive genomic instability, but in my opinion, I do not see any evidence that supports this claim. In fact, one might think that genomic instability is key to cell transformation.

We greatly appreciate your thorough review and insightful comments on our manuscript. We have taken your feedback seriously and have made several key revisions to address your concerns.

To your primary point about how ELF3 helps cells tolerate DNA damage, we have expanded our discussion to clarify the role of ELF3 in the context of BRCA1 deficiency and high replicative stress. We clarified that while ELF3 may not directly suppress excessive genomic instability, it plays a role in maintaining a balance that prevents catastrophic damage in BRCA1-deficient cells. Both BRCA1 deficiency and increased replication stress induce up-regulation of ELF3, which acts as a transcription factor, and it’s up-regulation leads to up-regulation of the expression of a variety of DNA replication-associated proteins that help to maintain homeostasis in the DNA replication process (Figure 5 E and F). Defects in ELF3 also do lead to disruption of the DNA replication process (Figure 5 G-I). While ELF3 cannot completely eliminate genomic instability, ELF3 essentially maintains genomic instability within a dangerous yet non-lethal range: higher than in normal cells, but not so high as to cause cell death.

This precarious balance can facilitate the transformation of LPs into a malignant state, as you pointed out.

In the revised manuscript, we emphasized that in cells with inherently low replicative stress, such as other non-LP mammary cells, the ELF3-associated mechanism might help cells endure the high replicative stress caused by BRCA1 deficiency without leading to cancerous changes. However, in LP cells, which naturally experience higher replicative stress, this ELF3-related mechanism may make them more susceptible to transformation into cancer cells. This supports our hypothesis that the combination of high replicative stress and BRCA1 deficiency specifically predisposes LP cells to tumorigenesis.

We have modified the working model to make it clearer.

**Reviewer #2 (Public Review):**
Summary:The manuscript focuses on a persistent question of why germline mutations in BRCA1 which impair homology-directed repair of DNA double-strand breaks predispose to primarily breast and ovarian cancers but not other tissues. The authors propose that replication stress is elevated in the luminal progenitor (LP) cells and apply the gene signature from Dreyer et al as a measure of replication stress in populations of cells selected by FACS previously (published by Lim et al.) and suggest an enrichment of replication stress among the LP cells. This is followed by single-cell RNA seq data from a small number of breast tissues from a small number of BRCA1 mutation carriers but the pathogenic variants are not listed. The authors perform an elegant analysis of the effects of BRCA1 knockdown in MCF10A cells, but these cells are not considered a model of LP cells.Overall, the manuscript suffers from significant gaps and leaps in logic among the datasets used. The connection to luminal progenitor cells is not adequately established because the models used are not representative of this population of cells. Therefore, the central hypothesis is not sufficiently justified.Strengths:The inducible knockdown of BRCA1 provided compelling data pointing to an upregulation of ELF3 in this setting as well as a small number of other genes. It would be useful to discuss the other genes for completeness and explain the logic for focusing on ELF3. Nonetheless, the connection with ELF 3 is reasonable. The authors provide significant data showing a role for ELF3 in breast epithelial cells and its role in cell survival.Weaknesses:The initial observations in primary breast cells have small sample sizes. The mutations in BRCA1 seem to be presumed to be all the same, but we know that pathogenic variants differ among individuals and range from missense mutations affecting interactions with one critical partner to large-scale truncations of the protein.The figure legends are missing critical details that make it difficult for the reader to evaluate the data. The data support the notion that ELF3 may participate in relieving replication stress, but does not appear to be limited to LP cells as proposed in the hypothesis.

We would like to sincerely thank you for your thorough review and constructive feedback on our manuscript. Your insightful comments and suggestions have been invaluable in guiding our revisions.

(1) Acknowledgment of Data Set Limitations and Additional Analyses: We fully acknowledge the importance of the concerns raised regarding the datasets used in our study. We have supplemented our manuscript with the missing information you pointed out and conducted additional analyses as suggested. These efforts have

(2) Challenges in LP Cell Experiments:

One of the most critical issues you raised was the lack of validation in LP cells, particularly concerning the role of ELF3 in these cells. We are acutely aware of the significance of this point. Following your review, we made extensive efforts to isolate and culture LP cells from both BRCA1-proficient and BRCA1-deficient patient samples. We tried various methods and invested substantial resources, including time, manpower, and materials, to establish a reliable protocol for isolating and cultivating LP cells in vitro. Unfortunately, despite our best efforts, we were unable to obtain a sufficient number of high-quality cells to generate solid and reproducible results.

The challenges we faced included the limited availability of patient tissues and the technical difficulties in consistently obtaining viable LP cells. Given the already extended timeline for the revision of this manuscript, we regretfully decided to forgo further attempts to perform these critical experiments with LP cells. In the revised manuscript, we have explicitly addressed the limitations of our cell models and provided a detailed discussion of the challenges faced in isolating LP cells. Despite these limitations, we believe that the consistency between our results and LP cell sequencing data provides valuable insights and a solid foundation for future studies.

(3) Data Presentation Improvements:

In response to your feedback, we have also made significant improvements to the data presentation in our manuscript. We updated and optimized figure legends and narrative sections to ensure that the data are clearly and accurately conveyed. These changes aim to enhance the readability and comprehensibility of our findings.

We greatly appreciate your valuable feedback, which has significantly contributed to the improvement of our manuscript. Your suggestions have helped us refine our arguments and present a more robust and nuanced interpretation of our data.

Thank you once again for your critical and constructive review. We look forward to your feedback on our revised manuscript.

**Recommendations for the authors:**

**Reviewer #1 (Recommendations For The Authors):**
As such, in addition to consolidating the role of ELF3 in promoting cell tolerance to replicative stress (or in suppressing genomic instability), I have a few comments the authors should consider to improve their manuscript.(1) I am not sure how cells have gained a growth advantage if they were arrested (Line 105-106). Perhaps the authors can elaborate.

Thanks for pointing this out and we are sorry for the misleading statement. We have revised the manuscript and would like to clarify that “survival advantage” may be more accurate than “growth advantage”, and since long-term DOX treatment led to decreased cell survival indicated by decreased number of colonies in Supplemental Fig. S1D, thus many cells died during DOX treatment. Therefore, the cells able to survive throughout DOX treatment and being collected for sequencing may have gained survival advantage compared to their counterparts who fail to survive.

(2) Figure 3D - From Western blotting of ELF3, forced expression of E2F6 does not appear to "block" HU-induced ELF3 up-regulation, but merely down-regulate basal level of ELF3, with the effect of HU still notable.

Thanks for the comment and we agree that E2F6 down-regulate ELF3 baseline expression levels and did not fully block ELF3 up-regulation. After calculating the foldchange after E2F6 overexpression, we did confirm that E2F6 overexpression still partially block HU-induced ELF3 up-regulation, with foldchange from 3.32 to 2.40, supporting our conclusion that HU-induced ELF3 upregulation is regulated by ATRChk1-E2F axis. It does, however, cannot be excluded that E2F6 also regulates ELF3 expression in other replication stress-independent ways, and we have revised the manuscript accordingly.

(3) Figure 3J & K - In my opinion, if BRCA1 knockdown were more efficient it remains formally possible that co-depletion of BRCA1 and GATA3 may exhibit additive effects in up-regulating ELF3 mRNA level.

Thank you for the comment. Actually, the BRCA1 knockdown efficiency in Figure 3J was shown in Supplemental Fig. S3B, and notably both BRCA1 and GATA3 knockdown were numerically more efficient in the double-knockdown group than in the single-knockdown group, individually. Thus, the higher ELF3 up-regulation in double-knockdown group in Figure 3J could be cause by the superior knockdown efficiency of both BRCA1 and GATA3. Nonetheless, we agree that it might be possible that BRCA1 and GATA3 still have separate functions in this experimental setting and marginal additive effect may exist, and the manuscript was revised accordingly.

(4) Figure 4 - Perhaps the authors can change its title to better summarise the findings. Cell sensitivity assays and xenograph experimentations may not necessarily relate to genomic instability.

Thank you for the great suggestion. To summarize the results more accurately, we have revised the title as “ELF3 can help cells tolerate replication stress and sustain cell survival”.

(5) Figure 5B&C - It would be important to document the time-dependent resolution of HU-induced DNA lesions by including additional time-points before, during, and after HU treatment.

We appreciate the suggestion to include additional time points to document the timedependent resolution of HU-induced DNA lesions. In our experiments, we observed that ELF3 knockdown leads to genomic instability both in the presence and absence of HU treatment. Specifically, Figure 5A and Figure S5 demonstrate that ELF3 knockdown increases genomic instability without HU treatment, indicating its role in maintaining genomic stability under normal conditions. On the other hand, Figure 5B, 5C, and 5D show that ELF3 knockdown under HU-induced replication stress further exacerbates genomic instability. This observation aligns with our finding that ELF3 expression increases in response to replication stress, suggesting its critical role in maintaining replication homeostasis under such conditions.

1. Figure 5F&I - Which ELF3 siRNA was used in these experimentations? Since the authors did not exclude off-target effects perhaps it may be worthwhile to include both ELF3 siRNAs for Panel F.

Thanks for your advice. The qPCR (Figure 5F) and DNA fiber assay (Figure 5I) were using siELF3-4 siRNA. And we repeat the qPCR experiments for Panel F using siELF3-5 siRNA (Supplement Fig. S5B).

We sincerely thank you for your thoughtful feedback and constructive suggestions. Addressing these points has strengthened our manuscript, and we are grateful for the opportunity to refine and clarify our work. We appreciate your critical evaluation and look forward to further constructive dialogue.

**Reviewer #2 (Recommendations For The Authors):**
(1) The data driving the hypothesis uses gene expression signatures as an indirect measure of replication stress. This is a critical concern.a. At this time, numerous gene expression signatures have been reported to be biomarkers of replication stress. Therefore, it would be valuable to apply additional gene expression signatures to examine the performance and the overlap in the results.The recent work by Takahashi et al., 2022 (https://pubmed.ncbi.nlm.nih.gov/36381660/) provides a signature that was derived independently and offers one that can be used to assess the performance of the signatures and stability of the conclusions.

Thank you for the valuable suggestion. We have done the replication stress evaluation of mammary cell subgroups using the Repstress score developed in the work you mentioned. The result showed that LP cells have trends of higher replication stress compared with other subgroups, though no statistical significance. This result, consistent with our previous analysis, indicated that LP cells have higher trends of replication stress levels. And we have added this data as the last line of Figure 1A in revised vision.

**Author response image 1. sa2fig1:** Replication stress pathway scores of different human normal mammary cell populations. The gene expression data were from Lim et al. (3).

b. A direct measure of replication stress in LP cells would be important to confirm the gene expression signature. Therefore, performing immunostaining for markers of replication stress (eg gamma-H2AX foci, DNA fiber assays) would provide more direct data to support the assertions.

Thank you for this suggestion and we totally agree that experiments revealing replication stress levels by investigating common markers, e.g., gamma-H2AX foci, DNA fiber assays, will provide vital evidence for our hypothesis. However, since our last response, we have been diligently trying to obtain LP cells for these experiments but encountered technical challenges while attempting to isolate and culture LP cells in vitro.

In the discussion part, we have revised the manuscript to emphasize that the data obtained from MCF10A should be interpreted with caution and there are certain gaps between the cell models and LP cells.

(2) The depth of single-cell sequencing can often be limiting. Therefore, a supplementary table listing the genes used for the replication stress signature and the frequency that they are observed in the single-cell sequencing data. This is needed to ensure that the replication stress score does not reflect a small subset of the replication stress signature genes.

Thanks very much for this evaluable suggestion. We have provided an expression matrix of genes for the replication stress signature in the revised version (Supplementary Table S1), And we also calculated the average expression level of each gene in the cells. As shown in Author response image 2, these genes expressed relatively low at the single-cell level (with counts≤10), The expression differences among genes are relatively small. Thus, we excluded the possibility that several high-expressed genes significantly affect the replicative stress score.

**Author response image 2. sa2fig2:** Average counts of Top 50 genes for the replication stress signature.

(3) As only 4 BRCA mutation carriers are analyzed, it is critical that the mutations be reported for these individuals because pathogenic variants differ in their effects and interactions with the DNA repair machinery in cells.

Thanks for the suggestion and the information of 4 BRCA1 mutant carriers were added in Supplemental Table S2.

(4) The figures throughout lack critical details making it difficult to evaluate. Figure 1A states that these are "replication stress pathway scores..." but there is no evaluation of levels of statistical differences. The heat map has what appears to be a log unit score between +2 and -2 but it is unclear whether it is log2 or log10 or some other unit. In 1B, the replication stress scores are visualized as relative values between 0 and 0.1, but there is no indication of what this means or whether there is a statistically significant difference in the levels among the populations. As tumors are composed of multiple cell types, it should be stated how the "tumor cells" are uniquely identified in the figure legend. The lack of critical information is common across many of the figures making review frustratingly difficult.

Thanks for the suggestion. We have added the statistical analysis and scale in Figure 1A legend. For Figure 1B, replication stress was calculated by sum of replication stress gene expression and presented as ln value. We have provided a quantitative figure and statistical tests (by Mann-Whitney) of replication stress scores for various cell types (Supplementary Figure 1A).

In addition, we added details of identification of tumor cells in the method section in the revised manuscript. Briefly, the adjacent normal breast sample served as a control to filter various types of normal cells from tumor samples. the normal cells from the tumor sample were merged with the same types of normal cells from adjacent normal breast samples, leaving one cell cluster only generalized by tumor sample. These tumor specific clusters were considered as malignant cell populations. We further found that the malignant cell population showed higher UMI counts than the normal cell populations, consistent with active metabolism in the malignant cells. More importantly, ER, PR, and HER2 expression of the malignant cells in each case were exactly matched with the clinical records. Finally, we utilized InferCNV to validate malignant cells subset as higher copy number alterations (CNAs) detected in the malignant cells compared with normal cells.

(5) The hypothesis states that the LP cells are uniquely sensitive to deficiency in BRCA1 compared to other cells. However, the authors use knockdown of BRCA1 in MCF10A cells which are generally considered to be basal cells and not LP cells.

Thanks for the comment. We totally agree that MCF10A cannot reflect the LP features and was mainly used as a normal mammary cell line model. We have tried to obtain human LP to perform some experiments but have all failed due to the cell vulnerability and difficult to be passed on in vitro. The gap between MCF10A and LP cells was stressed in the discussion part.

(6) Figure 2, the number of samples being compared is not listed for most of the panels. It appears that ELF3 is enriched in subsets of breast cancers, but much of the data is not focused on BRCA1-deficient tumors. Therefore, the data appears to show that ELF3 expression is more of a generalized feature of TNBCs (which has been reported previously) and dilutes the support for the hypothesis. Therefore, panels C-G raise concerns regarding the overall hypothesis that LP cells are the cell type that is affected.

Thanks for the suggestion. We have added the number of samples in Figure 2 legends.

Our analysis focus on basal subtype because of the well-known relationship between BRCA1 deficiency and this subtype. Our results demonstrate the association between ELF3 expression and basal, TNBC, as well as HER2+ subtype, consistent with previous reports. Since TNBC also has high replication stress levels *(NPJ Breast Cancer. 2020 Sep 7;6:40.)*, ELF3 upregulation in this subtype may not be solely due to BRCA1 deficiency, and we totally agree that this analysis may dilute the relationship between ELF3 and BRCA1. We have revised the discussion part to be more precise on this.

(7) Figure 3 provides experimental support for the hypothesis. While panel A is of interest, the legend lacks any description beyond "normal mammary tissue" and that there are non-carriers and carriers of BRCA1 mutations. Is this from bulk RNAseq data or single-cell RNAseq data? How many carriers and how many noncarriers? Panel E is ENCODE data from MCF7 cells that are ER+ luminal subtype so it is unclear if this is relevant to the LP cells that are the focus of the hypothesis.

Thanks for the comments. Figure 3 panel A was from single-cell RNAseq data, including 3 BRCA1 WT patients and 4 BRCA1 mutant patients. All cells (normal cells and tumor cells) are involving, and ELF3 expression was normalized by reads in each cell. We have added this information in the figure legend.

It has been difficult to obtain ENCODE data in LP cells. The effect of E2F1 on regulation of ELF3 was validated in MCF10A cells by experiment and consistent with MCF7 ENCODE data, thus we suggest this effect can be conserve in mammary cells, but further confirmation in LP cells is needed. We have revised the manuscript to note that.

(8) In Figure 4, the authors use BRCA1-deficient breast cancer cells to show the reliance on ELF3 and suggest that this is specific to this genetic lesion and not other subtypes. However, there is no data to show that this is not observed using ER+ cells or TNBC that are not BRCA1-deficient cell lines or models.

Thanks for pointing this out. As ELF3 knockdown in MCF10A resulted in increased genomic instability (Supplement Fig S5) and less capability to resolve replication stress (Figure 5B), we believe that ELF3 can help deal with replication stress not specifically in BRCA1-mutant cells, but also normal mammary cells, and also multiple cell lines with distinct backgrounds as suggested in Figure 4G, 4H and Supplement Fig S4G. The special link between ELF3 and BRCA1 is reflected by ELF3 significant upregulation upon BRCA1 deficiency, but not ELF3 downstream functions.

(9) Figure 5 provides the first direct evaluation of biomarkers of replication stress (gamma H2AX, 53BP1). DNA fiber assays provide the most direct evaluation of replication fork kinetics, and therefore, replication stress. The knockdown of BRCA1 and ELF3 appear to phenocopy one another in the HCC1937, but there is no other cell type to show whether this is specific for BRCA1-deficient cells. For example, the MCF7 cells show E2F1 binding to ELF3 (Figure 3E) and may show replication stress upon knockdown of ELF3. Without testing this, the authors cannot suggest that the effect is linked to BRCA1 status. The authors do not identify the BRCA1 mutation in these cells and whether there is homozygous loss. Similarly, the mutational status in the SUM149PT cells should also be stated. These need to be added to aid interpretation of the results.

Thank you for the constructive advice. We have added information regarding BRCA1 status of HCC1937 and SUM149PT. As discussed before, the results from Figure 4G and 4H suggest that ELF3 expression is associated with sensitivity to replicationstress-inducing-drugs across many cell lines. Thus ELF3 can maintain the stability of DNA replication is not specific to BRCA1-deficient cells. The reliance of ELF3 in BRCA1-deficiency we proposed is mainly focus on the fact that ELF3 is upregulated in BRCA1 deficient conditions, plus ELF3 may help cells tolerate replication stress during the transformation, therefore the resulted tumor cells-that is BRCA1-deficient breast cancer cells-may be more sensitive when losing ELF3 expression.

(10) While the data in Figure 6 are valuable extensions of the gene signature derived from the MCF10A cells with BRCA1 knockdown, only 2 BRCA1 carriers are reported. As carriers bear heterozygous mutations in BRCA1, haplo-insufficiency would be necessary to generate the signature. The authors do note the publication by Panthania et al, but there are relatively few examples of haploinsufficiency. It should be noted that Sedic et al., 2015 also suggested haploinsufficiency in breast epithelial cell cultures from BRCA1 heterozygotes which appears to cause premature senescence, possibly via replication stress. However, this was observed in the basal epithelial cells. Therefore, this appears to be a feature of the breast epithelium more generally and is not enriched or limited to the LP cells.

Thanks very much for your valuable suggestion. We have revised the discussion part to involve this important work and we fully agree that BRCA1 deficiency can cause replication stress not limited to LP cells. While in fact, the point we would like to address in Figure 6 is that BRCA1 deficiency modules the transcription profile towards LP-like cells, but not other-subtype-like cells, in normal mammary cells. We observed surprisingly similar profile between BRCA1-deficient cells and LP cells, suggesting there might be an inherent function of BRCA1 to mediate LP genes transcription. Furthermore, the data indicate that ELF3 has a tighter association with LP genes than other recognized LP-specific transcription factors like ELF5 and EHF, which are of the same family of ELF3. This result is intriguing since ELF3 can be upregulated by BRCA1 deficiency and replication stress. We assume that ELF3 could be a transcription node downstream of BRCA1 deficiency and modulate LP genes expression, and this process might be limited to LP cells since ELF3 has the highest expression levels in LP. Nonetheless, this hypothesis is also needed to be validated in LP cells by experiments.

We would like to express our deepest gratitude to the reviewers for their thorough and constructive feedback. Their insightful comments have been invaluable in guiding the revisions of our manuscript, helping us to clarify our hypotheses and strengthen the presentation of our findings. While we encountered some challenges, particularly with the isolation and culturing of LP cells, we made significant efforts to address the reviewers' concerns to the best of our ability. We have updated our manuscript accordingly, ensuring that all issues raised have been addressed comprehensively. We believe that these revisions have substantially improved the quality and clarity of our work, and we are excited to share our findings with the scientific community. Thank you once again for the opportunity to revise our manuscript, and we look forward to your feedback on the updated version.